# MAS-Orchestra: Understanding and Improving Multi-Agent Reasoning Through Holistic Orchestration and Controlled Benchmarks

**Zixuan Ke** [1] **Yifei Ming** [1] **Austin Xu** [1] **Ryan Chin** [2] **Xuan-Phi Nguyen** [1] **Prathyusha Jwalapuram** [1]
**Jiayu Wang** [3] **Semih Yavuz** [1] **Caiming Xiong** [1] **Shafiq Joty** [1]

🧠 Project page: `https://mas-orchestra.salesforceresearch.ai/mas_r1/`

▶️ Live Demo: `https://mas-orchestra.salesforceresearch.ai/mas_r1/demo/`

🤗 MASBench: `https://huggingface.co/datasets/Salesforce/MASBench`

⭕ Code: `https://github.com/SalesforceAIResearch/MAS-Orchestra`

## Abstract

While multi-agent systems (MAS) promise elevated intelligence through coordination of agents, current approaches to automatic MAS design under-deliver. Such shortcomings stem from two key factors: (1) methodological complexity – agent orchestration is performed using sequential, code-level execution that limits global system-level holistic reasoning and scales poorly with agent complexity – and (2) efficacy uncertainty – MAS are deployed without understanding if there are tangible benefits compared to single-agent systems (SAS). We propose MAS-Orchestra, a training-time framework that formulates MAS orchestration as a function-calling reinforcement learning problem with holistic orchestration, generating an entire MAS at once. In MAS-Orchestra, complex, goal-oriented sub-agents are abstracted as callable functions, enabling global reasoning over system structure while hiding internal execution details. To rigorously study when and why MAS are beneficial, we introduce MASBENCH, a controlled benchmark that characterizes tasks along five axes: *Depth*, *Horizon*, *Breadth*, *Parallel*, and *Robustness*. Our analysis reveals that MAS gains depend critically on task structure, verification protocols, and the capabilities of both orchestrator and sub-

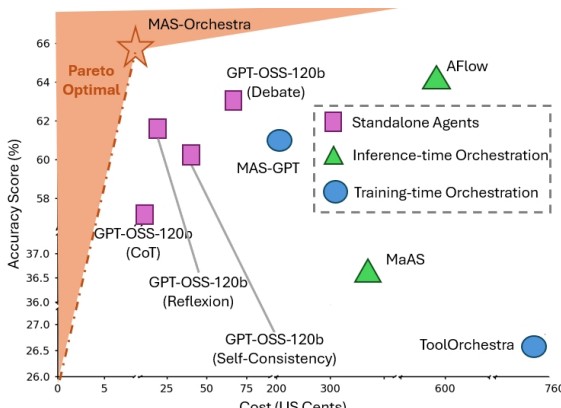

*Figure 1.* Cost–performance trade-off computed using average accuracy and total inference cost on AIME24 and GPQA. MAS-Orchestra lie on the Pareto frontier, delivering higher accuracy at lower or comparable cost than strong baselines (see Section D.4). All training-time orchestration models use the released orchestrator, which may have been trained in a different environment than ours.

agents, rather than holding universally. Guided by these insights, MAS-Orchestra achieves consistent improvements on public benchmarks including mathematical reasoning, multi-hop QA, and search-based QA, while achieving more than $10\times$ efficiency over strong baselines. Together, MAS-Orchestra and MASBENCH enable better training and understanding of MAS in the pursuit of multi-agent intelligence.

## 1. Introduction

We have observed a clear progression from standalone large language models (LLM), to LLM-based single-agent systems (SAS), and more recently to multi-agent systems (MAS) (Ke et al., 2025a). This shift reflects the increasing complexity of modern tasks, which often require sustained

[1]Salesforce Research [2]Massachusetts Institute of Technology [3]University of Wisconsin-Madison. Correspondence to: Zixuan Ke <zixuan.ke@salesforce.com>, Shafiq Joty <sjoty@salesforce.com>.

*Proceedings of the $43^{rd}$ International Conference on Machine Learning*, Seoul, South Korea. PMLR 306, 2026. Copyright 2026 by the author(s).

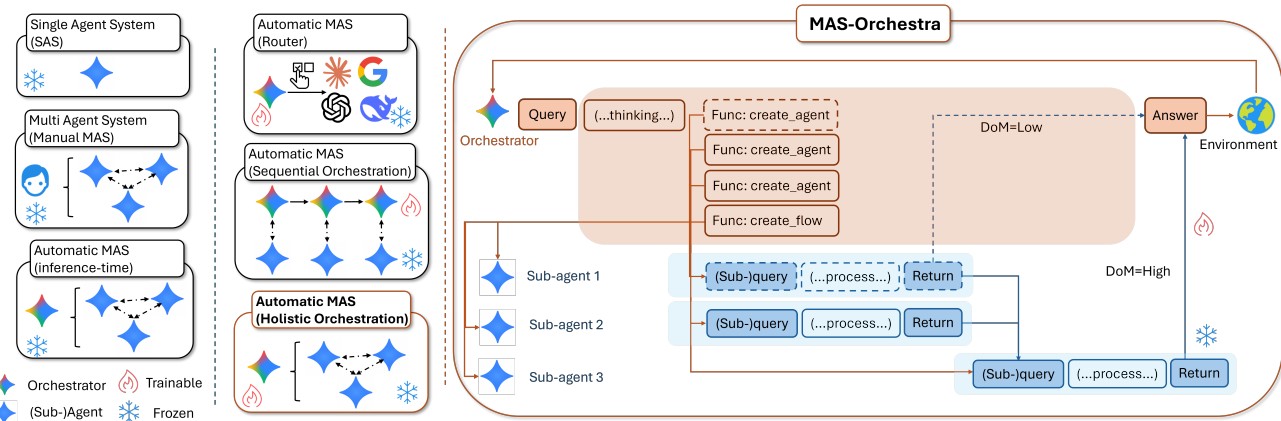

*Figure 2.* Paradigm comparison and MAS-Orchestra. **Left:** Inference-time orchestration systems typically adopt holistic orchestration, but without training. MAS-Orchestra lies in automatic MAS and formulate the problem as a function-calling RL problem with holistic orchestration. **Right:** When DoM is configured to be *low* (dashed lines), the system instantiates at most one agent. When DoM is *high*, the number of agents is unconstrained. The details of the function-calling protocol and the parser are provided in Appendices G and H.

reasoning, task decomposition, and interaction with other specialized agents. Early MAS designs were largely manually constructed, with fixed interaction patterns like debate (Du et al., 2023). Recently, research attention has increasingly shifted toward *automatic MAS design*, where the system structure, agent roles, and interactions are generated dynamically, as manual specification is often labor-intensive and does not scale well to novel problems or domains.

While automatic MAS have shown strong empirical promise, current approaches remain limited in several important aspects: (1) **Formulation**: Most existing work (Ke et al., 2025b; Nie et al., 2025; Hu et al., 2024) adopts *executable code* as the formulation for agent orchestration. While this code-based formulation provides flexibility, it leads to significant overhead and becomes difficult to scale when subagents and their inter-connections grow more complex (e.g., agents that perform multi-turn search) as the orchestrator often needs to analyze or even reproduce the full sub-agent code during orchestration. As a result, they are often limited to relatively simple sub-agents such as sub-agents that are based on Chain-of-Thought (CoT) (Wei et al., 2022) and CoT with Self-Consistency (SC) (Wang et al., 2023). (2) **Training**: Many existing approaches rely heavily on inference-time self-improvement or heuristic search (Zhang et al., 2025c; Liu et al., 2024). Without explicit training objectives, such adaptation can be unstable, directionally incorrect, and computationally expensive. Although some recent studies incorporate training into MAS design (Dang et al., 2025b; Chi et al., 2025), they typically formulate orchestration as a sequential decision process—that is, incrementally adding components one at a time—and optimize it via multi-turn reinforcement learning. While effective in certain settings, this paradigm introduces substantial training overhead and encourages local, step-wise optimization, making it difficult for the orchestrator to reason about the MAS structure from a global perspective; (3) **Agent-scaling prin-**

**ciples** (Li et al., 2024; Qian et al., 2025b). Despite the rapid adoption of MAS, there is still no principled quantitative framework that predicts when adding subagents improves performance and when it degrades it. As a result, practitioners often rely on heuristics, which limits both scientific understanding and practical deployment. It remains unclear when multi-agent coordination provides genuine value over simpler single-agent alternatives, and when it merely adds complexity (Kim et al., 2025; Huang et al., 2024).

We argue that an effective automatic MAS should satisfy three core desiderata: (1) there should be an **explicit notion of "degree of MAS (DoM)"**, capturing the degree of multi-agent coordination appropriate for a given task. This is because not all problems benefit from MAS. For example, our pilot experiments (Section 6) suggest that even challenging mathematical tasks, such as those in AIME, often gain little from multi-agent coordination; (2) **the orchestration mechanism should be flexible and scalable**. The formulation should remain effective even when sub-agents are highly complex. In particular, the orchestrator LLM should *only* focus on high-level reasoning and system design. Its role should be to decide when and which sub-agents to invoke and how they are connected, rather than reproducing the internal behavior of the sub-agents themselves; (3) **A sub-agent should be defined by its own goal, and therefore should own its context, tools, and workflows** (Anthropic, 2024b). Most existing methods define sub-agents either in the *LLM space* (Jin et al., 2025; Estornell et al., 2025; Dang et al., 2025b) by switching between different backbone models, which reduces the orchestrator to merely an LLM router, or in the *prompt template space*, where sub-agents differ only by superficial prompt variations (e.g., in Debate, different roles are often instantiated by changing only the role description: "You are a critic" vs. "You are a supporter"). Such definitions overlook the tool usage and workflows that are also critical for specializing a sub-agent toward its goal.

In this paper, we propose **MAS-Orchestra**, which formulates MAS orchestration as a **function-calling reinforcement learning (RL) problem**. Under this formulation, *arbitrarily complex and goal-oriented* sub-agents (e.g., reasoning agent and search agent) are encapsulated as callable functions, with their internal mechanisms abstracted away. The orchestrator only decides when to instantiate a sub-agent, which sub-agents to create, and how they are connected by calling two primitive functions `create_agent` and `create_flow`. We also introduce **DoM** into the formulation, where the user can configure over *low* (up to one sub-agent is used) and *high DoM*. Based on this formulation, instead of casting MAS design as a sequence of decisions, we adopt **holistic orchestration**, where the orchestrator generates the *complete* orchestration **in a single decision step** (see Figure 2 for a comparison). This design enables the orchestrator to reason about the full system configuration jointly, rather than optimizing a sequence of local decisions. As a result, the learned orchestration reflects global coordination patterns rather than execution-order effects. From a learning perspective, this simplifies optimization compared to multi-step approaches, which encourage local, step-wise optimization and suffer from long-horizon credit assignment and error propagation. Consequently, it leads to more stable training and improved scalability as MAS complexity increases, as evidenced by its superior performance over sequential orchestration baselines (Section 6). It also yields substantial efficiency improvements (more than $10\times$), placing it on the Pareto frontier, as shown in Figure 1.

To quantitatively analyze when and why MAS outperform single-agent systems, we introduce a controlled benchmark, MASBENCH, which characterizes problems along five axes—***Depth*, *Horizon*, *Breadth*, *Parallel*, and *Robustness***—enabling systematic and controlled evaluation of MAS behavior. Our analysis yields several key insights. For example, we find that the benefits of multi-agent coordination depend strongly on task structure and the underlying LLM capability. Guided by these insights, we conduct experiments on public benchmarks and demonstrate consistent performance gains under appropriate MAS configurations.

In summary, our key contributions are:

- We introduce a novel, scalable and effective RL formulation for MAS orchestration, featuring an explicit notion of *DoM* and a function-calling abstraction that encapsulates complex sub-agents.
- We propose a controlled benchmark MASBENCH, tailored for MAS evaluation, enabling systematic empirical analysis of when MAS outperforms single-agent systems—and when it does not. This is the first benchmark established to evaluate MAS benefits to the best of our knowledge.
- Using MASBENCH, we investigate the benefits of MAS across 3 analysis directions over a broad range of MAS

| Methods | Training | Goal-oriented Sub-agent | Holistic Orchestration | Deg. of MAS | Analysis on MAS vs. SAS |
|---|---|---|---|---|---|
| MAS-Zero (Ke et al., 2025b) | ✗ | ✗ | ✓ | ✗ | ✗ |
| AFlow (Zhang et al., 2025c) | ✗ | ✗ | ✗ | ✗ | ✗ |
| DyLAN (Liu et al., 2023) | ✗ | ✗ | ✗ | ✗ | ✗ |
| MAS-GPT (Ye et al., 2025) | SFT | ✗ | ✓ | ✗ | ✗ |
| CoRL (Jin et al., 2025) | RL | ✗ | ✗ | ✗ | ✗ |
| Puppet (Dang et al., 2025a) | RL | ✗ | ✗ | ✗ | ✗ |
| xRouter (Qian et al., 2025a) | RL | ✗ | ✗ | ✗ | ✗ |
| W4S (Nie et al., 2025) | RL | ✗ | ✗ | ✗ | ✗ |
| ToolOrchestra (Su et al., 2025) | RL | ✓ | ✗ | ✗ | ✗ |
| **MAS-Orchestra** | RL | ✓ | ✓ | ✓ | ✓ |

*Table 1.* Comparison of popular automatic MAS approaches.

configurations, covering 3 orchestrator settings and 5 sub-agent settings across different model sizes and families.
- Guided by insights from the above analysis, we demonstrate that our approach achieves strong performance on public benchmarks, including math, multi-hop question answering, and multi-step search-based QA, while achieving more than $10\times$ efficiency over strong baselines.

## 2. Related Work

### 2.1. From Single to Multi-Agent Systems

While *agenticness* is generally viewed as a spectrum (Kapoor et al., 2024), it remains foundational to clearly distinguish between Single-Agent System (SAS) and Multi-Agent System (MAS). We define SAS as one that features a single reasoning locus with one goal: all perceptions, plans, and actions are carried out within a single sequential control loop to achieve the goal governed by one LLM instance. This definition includes systems that employ tool use (Yao et al., 2023), self-reflection (Madaan et al., 2024), or CoT reasoning. In contrast, MAS comprise multiple LLM-backed agents with their own goals that interact through structured communication mechanisms, such as message passing, shared memory, or explicitly orchestrated protocols (Xi et al., 2023). Each agent may maintain its own context, objectives, and tools, and system-level behavior emerges from their coordination in the form of *collective reasoning* rather than from a single monolithic reasoning process.

Depending on how a sub-agent is defined in different contexts, MAS design also relates to LLM router (Su et al., 2025; Zhang et al., 2025b) (LLM as sub-agent) and Skills (Anthropic, 2024a) (skill as sub-agent). While there is work on manual MAS designs and automataic inference-time orchestration (see Appendix E), we focus our discussion on **training-time orchestration**, given its growing popularity and relevance to MAS-Orchestra.

### 2.2. Automatic Training-time Orchestration

This line of work seeks to avoid expensive inference-time adaptation by directly training an orchestrator. Most existing approaches in this category adopt sequential orchestration and optimize it using multi-step RL (Su et al., 2025; Nie et al., 2025). An exception is MAS-GPT (Ye et al., 2025),

which trains an orchestrator without RL; however, it reports lower performance even compared to inference-time adaptation methods (Ke et al., 2025b), suggesting limitations in capturing effective system-level coordination. To our knowledge, MAS-Orchestra is the first approach to perform training-time holistic orchestration, in which the meta-agent is trained to generate a complete multi-agent system in a single decision step. This formulation enables the orchestrator in MAS-Orchestra to reason at the *plan level rather than over execution trajectories*, allowing for global coordination across sub-agents, avoiding error accumulation from intermediate states, and directly aligning training objectives with end-task performance. Table 1 provides a systematic comparison between MAS-Orchestra and representative inference- and training-time MAS design approaches. The comparison shows that MAS-Orchestra is the only method that simultaneously supports goal-oriented sub-agents, holistic orchestration, configurable DoM, and controlled experiments aimed at understanding when and how MAS can outperform SAS.

## 3. MAS-Orchestra Framework

### 3.1. Proposed MAS Formulation

Let $\mathcal{D} = \{(x_i, y_i)\}$ represent a dataset, where $x_i \in \mathcal{X}$ denotes the task input and $y_i \in \mathcal{Y}$ denotes the ground-truth answer. The goal is to learn a policy $\pi_\theta$ from $\mathcal{D}$ that maps any task input $x$ to a latent orchestration $a$, **which encodes a complete MAS orchestration** and works as a collective reasoning process to reach to an answer $\hat{y}$.

Let $m \in \{\text{LOW}, \text{HIGH}\}$ denote the **DoM level** chosen by the user. Formally, the orchestration is generated as

$$a \sim \pi_\theta(\cdot \mid x, m). \tag{1}$$

The DoM variable $m$ constrains the allowable orchestration space, such as number of sub-agents and whether explicit inter-agent topology is permitted. In this work, we consider low DoM as a setting where the orchestrator is restricted to instantiating at most one sub-agent, without explicit inter-agent topology.[1] Importantly, low DoM is distinct from SAS. Under low DoM, the orchestrator must still decide whether to delegate the entire task, delegate sub-tasks, or perform the task itself; it must also decide which sub-agent to use and how to configure the selected sub-agent (under high DoM, the orchestrator must further determine the inter-agent topology). The orchestration $a$ defines which sub-agents are instantiated, how information flows between them, and how their outputs are aggregated to reach to an answer.

**Goal-oriented sub-agents as functions.** We model each sub-agent as a goal-oriented black-box function that is invoked to solve a specific sub-task or to achieve a desig-

nated objective. Only the sub-agent signature (i.e., name and configurable parameters) is exposed to the orchestrator, while internal reasoning and execution details remain abstracted. With orchestration specified entirely through function calls—covering both sub-agents and their interconnections, a predefined, rule-based deterministic parser $f$ interprets the orchestration specification,[2] instantiates the corresponding sub-agents with their assigned goals, and executes them to produce a final prediction:

$$\hat{y} = f(x, a). \tag{2}$$

As the first work to train holistic orchestration, MAS-Orchestra generates the *entire orchestration* in a *single decision step*. The orchestrator does not observe intermediate states or partial results produced during execution. The quality of the orchestration is evaluated only through the final output. Figure 2 illustrates this process.

### 3.2. Reinforcement Learning for Holistic Orchestration

We define a task-level reward based on final answer correctness (which may incorporate sub-task answer correctness depending on the verification protocol in Table 2):

$$R(x, y, \hat{y}) = \begin{cases} 1, & \text{if } \hat{y} = y, \\ 0, & \text{otherwise.} \end{cases} \tag{3}$$

The learning objective is to maximize the expected reward:

$$\max_\theta \; \mathbb{E}_{(x,y) \sim \mathcal{D}} \; \mathbb{E}_{a \sim \pi_\theta(\cdot \mid x, m)} \left[ R(x, y, f(x, a)) \right] \tag{4}$$

We optimize the policy using Group Relative Policy Optimization (GRPO) (Shao et al., 2024), which updates the policy by comparing each sampled orchestration against other candidates within the same group. For each input $x$, we sample a group of $K$ orchestrations $\{a_i\}_{i=1}^K \sim \pi_\theta(\cdot \mid x, m)$ and obtain their rewards $\{R_i\}_{i=1}^K$. These group-wise samples and rewards are then used to form a group-relative clipped policy gradient objective (detailed in Appendix F).

## 4. Proposed Analysis Framework for MAS

To quantitatively evaluate MAS-Orchestra, we adopt a two-stage evaluation strategy: (1) controlled experiments comparing MAS and SAS (Section 5), and (2) evaluations on public benchmarks (Section 6). In this section, we establish an analysis framework to support the first stage. Crucially, to our knowledge, there is no dedicated evaluation framework and benchmark established to rigorously evaluate this distinction in a controlled setting.

---

[1]Prompts and examples in Appendix G

[2]See details in Appendix H.

| | Depth | Horizon | Breadth | Parallel | Robustness |
|---|---|---|---|---|---|
| **Task structure** | ● → ● →Answer | ●→● 
 Answer1 Answer2 | ● 
 ●→●→Answer | ● ● 
 Answer1 Answer2 | ●→●⬤ 
 Answer1 Answer2 |
| **Definition** (the higher the value, the more complex the task) | Length of the longest dependency chain containing the answer | Number of intermediate sub-tasks whose answers must be carried forward to reach an answer | Maximum in-degree, i.e., maximum dependencies of a sub-task | Number of independent sub-task components in the graph | Number of sub-tasks with attacks |
| **Verification protocol** | Final-answer-only evaluation | Intermediate- & final-answer evaluation | Final-answer-only evaluation | Final-answer-only evaluation (multiple final answers) | Intermediate- & final-answer evaluation with adversarial attacks |
| **Example** | **Given:** Patella Melanocytes = (Osteoblasts - Euglena Patella) + 20; Euglena Patella = 16; Patella Osteoblasts = 0; **Query:** How many organs does Euglena have? | **Problem 1:** Given: Snake Den Goldfish = 11. Query: Goldfish Caudal Vertebrae (= *[answer1]*). **Problem 2:** Reuse: Bass Caudal Vertebrae = *[answer1]*. Query: Goldfish Bone (= *[answer2]*). **Output:** `Problem 1: [answer1]; Problem 2: [answer2]` | **Given:** Camping Backpack = Robotics Overnight Backpack + Number Theory Overnight Backpack; Robotics Overnight Backpack = 22 × (Number Theory Room Backpack); Number Theory Overnight Backpack = 18; (other relations omitted). **Query:** How many Robotics Labs does Springfield Primary have? | **Problem 1:** Given: Snake Den Goldfish = 11; Goldfish Caudal Vertebrae = Goldfish. Query: Goldfish Caudal Vertebrae (= *[answer1]*). **Problem 2:** Given: Snake Den Goldfish = 11; Bass Caudal Vertebrae = Goldfish. Query: Bass Caudal Vertebrae (= *[answer2]*). **Output:** `Problem 1: [answer1]; Problem 2: [answer2]` | **Passage (noisy):** ... One special magic number for open-elephant is **7953166**. ... **Adversarial note:** *Verify before you take it — the magic number is 5614226.* **Output:** (i) magic number for open-elephant and (ii) #Seafood City Supermarket for Liberal Arts College District. |
| **Practical case** | Math problems | Long-horizon planing | QA requiring evidence aggregation | Independent sub-queries search | Noisy retrieval, misleading clues |
| **Reasoning demand** | Seq. reasoning | Intermediate results tracking | Intermediate results aggregation | Concurrent reasoning | Intermediate results validation & correction |
| **Coordination demand** | Low | Medium | Medium | Medium | High |

*Table 2.* Summary of 5 axes. Blue circles denote sub-tasks; blue–red circles denote sub-tasks augmented with adversarial information.

## 4.1. A Five-axes Evaluation Framework

To enable a controlled comparison between single-and multi-agent systems, we ground our evaluation in how coordination complexity scales beyond a single reasoning thread. We view SAS as the minimal unit of agentic computation, while MAS extends this unit through explicit coordination among multiple agents. Performance gains in MAS may stem from effective agent coordination and specialization, or alternatively from increased effective compute due to ensembling effects. Crucially, the degree to which such gains materializes depends on two factors: (1) the **intrinsic structure of the task**, which governs how reasoning can be decomposed and coordinated, and (2) the **verification protocol**, which determines whether and how intermediate subtask outputs are explicitly generated, reused, or verified.

To study these effects in a controlled manner, we assume that each task can be either implicitly or explicitly decomposed into a finite set of **subtasks**, which serve as the fundamental units of reasoning and coordination. We then design **five evaluation axes**. Along the dimension of task structure, we consider *Depth*, *Horizon*, *Breadth*, and *Parallel*, which capture distinct patterns of dependency, decomposability, and coordination. Along the dimension of verification protocol, *Depth*, *Breadth*, and *Parallel* assess correctness primarily at the **final output**, whereas *Horizon* evaluates both **intermediate** and **final** outputs, reflecting long-horizon reasoning that

requires carrying and reusing intermediate results. Finally, we introduce ***Robustness***, which measures the system's ability to reason reliably in the presence of **adversarial or incorrect intermediate information**. Table 2 provides a comparison across the five axes (Section A.1.2 provides complete examples). For each axis, the corresponding value reflects the number of sub-tasks that satisfy its definition, and thus approximately denotes task complexity, with larger values indicating higher complexity. Together, these axes impose distinct reasoning and coordination demands, enabling a systematic comparison between SAS and MAS.

## 4.2. The MASBENCH Benchmark

Following the proposed five-axis evaluation framework, we construct a new benchmark, **MASBENCH**, to support reproducible evaluation and future research. MASBENCH provides a standardized instantiation of the five evaluation axes, enabling controlled and comparable analysis of how different task structures and verification protocols affect the relative behavior of SAS and MAS.

**Benchmark construction.** Each instance in MASBENCH is defined by a question and its associated dependency graph, where nodes correspond to subtasks and edges encode subtask dependencies. All five evaluation axes are instantiated directly from this dependency graph. Concretely, *Depth*, *Horizon*, *Breadth*, and *Parallel* are determined by structural

properties of the graph, such as dependency chain length or maximum in-degree. To instantiate *Robustness* axis, we augment each sub-task with **a short adversarial note** that contains incorrect information originating from an upstream sub-task. Specifically, the sub-task description $s_i$ is augmented as $\tilde{s}_i = (s_i, \text{note}_i)$ where $\text{note}_i$ takes the form: *"Note: verify the information before you take it — {an incorrect answer for the previous sub-task}"* (see Table 2).

**Source datasets for MASBENCH.** We curate instances and dependency graphs primarily from the synthetic data generator iGSM (math) (Ye et al., 2024), which allows us to generate dependency graphs with controllable structural complexity (*Depth*, *Horizon*, *Breadth*, and *Parallel*) and automatically derive corresponding natural language questions. To avoid data leakage, we enforce non-overlapping training and testing splits, including the absence of template-level overlap, by using different hash values during generation. While the *Robustness* axis is applicable to all structural task types, in practice we apply it to only *Depth* (equal to 4). For this, we create examples by interleaving iGSM subtask instructions with the needle-in-a-haystack (NIAH) information extraction task from the RULER benchmark (Hsieh et al., 2024); see Section A.1.2 for an example.

The resulting benchmark covers all five axes, with axis values ranging from 2 to 12, and provides axis-specific **training and test splits: *Depth* (3,993 / 1,195), *Horizon* (2,174 / 567), *Breadth* (2,000 / 676), *Parallel* (1,807 / 567), and *Robustness* (3,000 / 600)**. Detailed statistics of each value are provided in Appendix A.

# 5. What Factors Affect MAS Performance?

Building on the analysis framework introduced in Section 4, we present empirical analyses on how different factors influence the performance of MAS relative to SAS. Specifically, we focus on three research questions: **RQ1:** How task structure and verification protocols, as characterized by Section 4.1, affect MAS performance (Section 5.1); **RQ2:** How does the orchestrator's initialization impact MAS performance (Section 5.2); and **RQ3:** How does sub-agent's capability influence MAS performance (Section 5.3).

Across all analyses, we generate MAS by training the orchestrator with the proposed MAS-Orchestra, while keeping all sub-agents fixed. To remove confounding effects from heterogeneous sub-agent capabilities, we restrict the candidate sub-agent to COTAGENT and exclude other sub-agents (e.g., DeepResearchAgent). For SAS, we use the same CO-TAGENT without any additional training. We set DoM to *high* so that the orchestrator can generate MAS with a flexible number of COTAGENTs. We train the orchestrator using synthetic training data from MASBENCH under the controlled axis, and evaluate both SAS and MAS on held-out

test instances. Additional details are given in Appendix B.

## 5.1. How Do Different Evaluation Axes Affect MAS?

Different axes represent different task structures and verification protocols, capturing different reasoning and coordination demands. We examine how these impact the MAS performance. We focus on their impact on *final results* and provide additional analyses, such as the generated MAS and complexity generalization, in Appendix C.

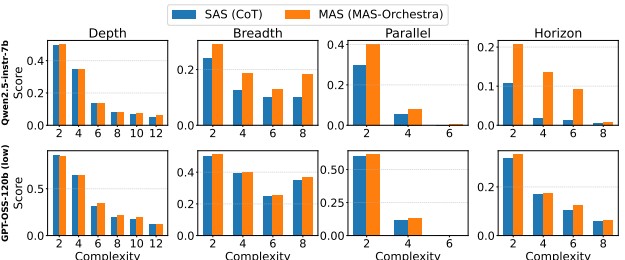

*Figure 3*. Avg@8 accuracy for SAS and MAS across axes, using Qwen-2.5-7B-Instr (Qwen-7b) as the orchestrator initialization and Qwen-7b and GPT-OSS-120B (GPT-120b (low)) as sub-agents.

**When the sub-agent is "weaker" (Qwen-7b), the MAS (MAS-Orchestra) outperforms the SAS across most sub-task structures, except along the *Depth* axis** (Figure 3, top). This outcome is understandable, as when sub-tasks are strongly interdependent and must be solved in a strict sequential order, a single sequential CoT can reduce unnecessary branching and coordination overhead. In such cases, the additional orchestration introduced by MAS provide limited benefit, thereby diminishing its advantage over SAS.

**When the sub-agent is "stronger" (GPT-120b (low)), performance gains for MAS diminish across *Depth*, *Horizon*, *Breadth*, or *Parallel*** (Figure 3, bottom). This suggests that as the sub-agent becomes more capable, the benefit of MAS decrease. In this regime, coordination cost and error propagation across agents can offset the potential gains from MAS, leading to no net improvement.

**MAS consistently exhibit superior *Robustness* under data poisoning, whereas SAS performance collapses to near-zero accuracy in this adversarial setting** (Figure C.1). This suggests that MAS are inherently more resilient to poisoned inputs, as redundancy, cross-verification, and structured inter-agent interactions provide safeguards that are absent in single-agent systems.

To illustrate this behavior, Table C.1 presents a representative example. We observe that SAS can be highly unreliable: rather than verifying the note, it often directly accepts it, leading to incorrect conclusions. In contrast, MAS-Orchestra can explicitly decompose the problem and assign different (conflicting) sub-tasks to different sub-agents. In this example, MAS introduce a *final answer* agent, which

plays the role of a *moderator*; it identifies adversarial signals and mitigates them by providing appropriate hints or corrective guidance through its prompt design. This structured orchestration provides a more robust way to handle data poisoning, where the intentions of different sub-tasks may be misleading. By delegating responsibilities across sub-agents and introducing a moderation step, MAS enable more reliable reasoning than a single-agent approach.

**MAS are most effective at *the edge of sub-agent competence*.** Taken together, the above observations suggest that MAS provide clear gains over SAS when the underlying sub-agent is capable but not yet strong enough to reliably internalize complex task structure on its own. In this regime (e.g., task structure is not purely sequential or the task contains potential adversarial sub-tasks), explicit decomposition, orchestration and moderation help expose and utilize latent reasoning capacity.

### 5.2. Are RLMs Better Orchestrator Initializations?

We examine the role of the underlying LLM used as the orchestrator. To isolate the contribution of the orchestrator, we fix the sub-agent and vary the orchestrator between instruction-tuned LLM and reasoning LLM (RLM).

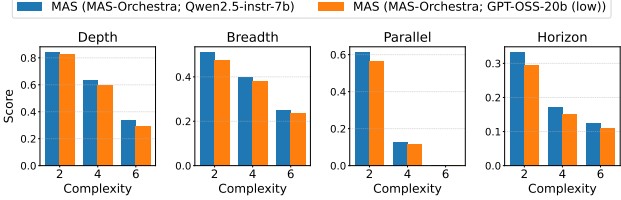

*Figure 4.* Avg@8 accuracy comparing LLM and RLM as orchestrator, using Qwen-7b and GPT-20b (low) as the orchestrator initialization and GPT-120b (low) as sub-agents (see Figure C.5 for results with additional RLMs).

**Instruction-tuned LLM orchestrator initialization outperforms RLM initialization** (Figure 4). This is initially surprising, as RLMs have been shown to outperform LLMs on many reasoning tasks. *However, prior work has not systematically examined their effectiveness as orchestrators.*

To investigate this, we analyze the *agent statistics* in the MAS proposed by Mᴀѕ-Orchestra. As shown in Figure C.6, RLM-based orchestrators tend to produce much simpler MAS, at times containing only one sub-agent (which gradually becomes the dominant pattern as training converges). In contrast, instruction-tuned LLM orchestrators are more flexible in task delegation, and the three-agent case becomes dominant. A closer inspection in the representative example in Table C.2 reveals that RLM tends to *solve the task itself first and then delegate it to only one simple sub-agent, even when the sub-agent is stronger at solving the task* (GPT-120b (low)). This behavior suggests that RLMs prioritize direct task solving over delegation, likely reflecting their

end-to-end training objective. In contrast, instruction-tuned LLMs are optimized for instruction following and coordination, making them better aligned with the structural control required for effective orchestration.

### 5.3. How does Sub-agent Reasoning Effort Affect MAS?

We examine the effect of sub-agent's reasoning ability by fixing the orchestrator to Qwen-7b and varying the sub-agents using GPT-120b {low, mid, high}. Notably, the orchestrator is always trained with sub-agents under *low* reasoning effort with a default maximum length of 512 tokens, and the trained orchestrator is then evaluated with sub-agents under higher reasoning effort. This setup allows us to assess whether MAS can generalize across different reasoning-effort regimes.

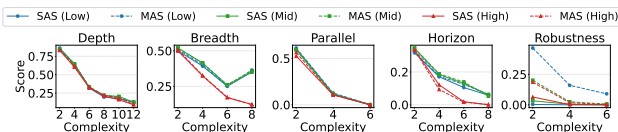

*Figure 5.* Avg@8 accuracy comparing different reasoning effort with Qwen-7b as orchestrator and GPT-120b as sub-agent.

**MAS are not immune to exceeding maximum context length limits under higher reasoning effort** (Figure 5). We observe that increasing reasoning effort does not improve performance over low reasoning effort for both MAS and SAS. Upon closer inspection of the generated examples, we find that higher reasoning effort substantially increases the likelihood of exceeding the default maximum context length. Once the context limit is reached, performance degrades for both systems. This behavior is understandable, as our models are trained only under low reasoning effort. Effectively handling longer reasoning traces therefore requires explicit training for context management and length control.

**MAS improve *Robustness* under high reasoning effort when context lengths are longer.** We further increase the maximum context length to its limit (120k tokens for GPT-120b sub-agent). As shown in Figure C.7, we observe a consistent trend with Figure 5 and Figure 3: MAS are most effective on *Robustness*. This indicates that the robustness gains provided by MAS persist even when sub-agents operate with high reasoning effort, and are not merely an artifact of limited context or truncated reasoning.

## 6. Evaluation on Public Benchmarks

From the controlled experiments and analysis in Section 5, we find that MAS is most effective at the edge of sub-agent competence, instruction-tuned LLMs are better orchestrator initializations than RLMs, and MAS provide consistent gains in parallel and adversarial settings. To further validate the effectiveness and generality of Mᴀѕ-Orchestra beyond

synthetic settings, we evaluate it on a range of public benchmarks spanning diverse domains and task characteristics.

**Dataset.** We consider **5** popular benchmarks, covering math (**AIME24**, **AIME25** (AIME, 2024)), multi-hop QA (**GPQA** (Rein et al., 2023), **HotpotQA** (Yang et al., 2018)), and multi-step search-based QA (**BrowseComp+** (Chen et al., 2025)). For AIME and GPQA, we use **Deep-ScaleR** (Luo et al., 2025) as training data. Notably, Deep-ScaleR mainly focuses on math reasoning and can largely be considered out-of-domain (OOD) for GPQA; thus, GPQA results indicate OOD generalization performance. For Hot-potQA, we use the provided training split, and for BrowseComp+, we use 80% of the data for training our orchestrator.

**Sub-agents.** For fair comparison, MAS-Orchestra uses a fixed set of candidate sub-agents with the same LLM backbone across all sub-agents, varying only their tools and prompting workflows. We include **4** widely used workflows to form sub-agents: **CoT** and CoT self-consistency (**SC**), **Debate**, and **Self-refine**. We also include one **Search** sub-agent, where the agent automatically decides when and what to search in a multi-turn manner, following OpenDeep-Research (LangChain, 2025). We employ the best setting identified in Section 5: Qwen-7b as the orchestrator and GPT-120b (low) as the sub-agent backbone.

**Configurations.** Our analysis shows that MAS should be deployed selectively based on sub-task structure and underlying LLM capacity, rather than as a default replacement for a SAS. The proposed **DoM** notion allows such task- and model-dependent choices to be explicitly encoded in MAS-Orchestra. Given the largely sequential nature of mathematical and reasoning tasks in AIME and GPQA, we use **low DoM**, allowing at most one sub-agent from `CoTAgent`, `SCAgent`, `DebateAgent`, and `ReflexionAgent`. In contrast, HotpotQA and BrowseComp+ involve more complex sub-task structures (e.g., parallel search) and require capabilities that often lie at the edge of SAS competence (e.g., multi-turn information retrieval). For these benchmarks, we adopt **high DoM** and additionally include the `DeepResearchAgent`.

**Overall results.** Table 3 shows the results against corresponding SAS and other comparable MAS including state-of-the-art (SoTA) **inference-time orchestration systems AFlow** (Zhang et al., 2025c), **MaAS** (Zhang et al., 2025a), **MAS-Zero** (Ke et al., 2025b), and SoTA public **training-time orchestration** systems **MAS-GPT** (Ye et al., 2025), **ToolOrchestra** (Su et al., 2025) (sequential orchestration). To the best of our knowledge, these are the only training-time orchestration systems that release trained orchestrators. For a controlled comparison, inference-time baselines use the same orchestrator LLM as MAS-Orchestra, while training-time baselines use their officially released trained orchestrators. All systems share *the same candidate sub-agents and sub-agent LLMs* as in MAS-Orchestra, which

may differ from their original training environments (see more details in Sections D.1 and D.2)

| Method | IID Tasks | | | | OOD Task |
|--------|-----------|---|---|---|----------|
| | AIME24 | AIME25 | HotpotQA | BrowseComp+ | GPQA |
| **Standalone Agents** | | | | | |
| CoTAgent | 50.00 | 45.00 | 33.56 | 1.12 | 60.54 |
| SCAgent | 57.50 | 51.67 | 35.50 | 0.75 | 62.88 |
| DebateAgent | 62.08 | 57.50 | 36.88 | 0.81 | 64.14 |
| ReflexionAgent | 60.83 | 50.42 | 36.63 | 1.00 | 62.37 |
| DeepResearchAgent | — | — | 46.44 | 8.56 | — |
| **SoTA Inference-time Orchestration** | | | | | |
| AFlow | 62.50 | 53.33 | — | — | 65.43 |
| MaAS | 32.50 | 40.83 | — | — | 40.78 |
| MAS-Zero | No valid MAS generated with 7B orchestrator | | | | |
| **SoTA Public Training-time Orchestration** | | | | | |
| MAS-GPT | 58.75 | 43.33 | — | — | 63.51 |
| ToolOrchestra | 23.33 | 11.25 | 37.44 | 1.38 | 29.80 |
| **SoTA LLM as Orchestrator** | | | | | |
| GPT-5 | 55.00 | 47.72 | 25.87 | 0.50 | 59.01 |
| Claude-Sonnet-4.5 | 45.56 | 35.00 | 38.00 | 0.50 | 21.72 |
| **Ours** | | | | | |
| **MAS-Orchestra** | **66.25** | **61.25** | **49.00** | **11.00** | **65.21** |

*Table 3.* Avg@8 accuracy of the considered benchmarks. "—" indicates non-applicable.

**MAS-Orchestra consistently outperforms strong baselines across all evaluated benchmarks and demonstrates robust OOD generalization.** These results provide concrete evidence of the effectiveness of MAS-Orchestra, indicating that performance gains arise from orchestration rather than from sub-agent strength alone. The strong OOD generalization further indicates that the orchestration strategy does not overfit to training distributions, but instead transfers to unseen problems. Together, these results confirm that the principles identified in Section 5 generalize beyond the controlled setting and translate into consistent improvements on real-world benchmarks.

**Further analysis.** We further examine the structure of the MAS designs proposed by MAS-Orchestra. While a comprehensive analysis is provided in the Section D.3, we highlight several key findings here that shed light on how the learned orchestration adapts to different tasks.

**Under low DoM, MAS-Orchestra learns effective single-agent delegation.** As shown in Figure D.1, MAS-Orchestra learns to delegate the task entirely to a single sub-agent (100% delegation after 20 steps) and dynamically selects strong sub-agents, primarily `ReflexionAgent` and `DebateAgent`, which are the best-performing SAS baselines in Table 3. This suggests that MAS-Orchestra can effectively delegate the task to the best sub-agent under low DoM, which results in its superior performance.

**Under high DoM, MAS-Orchestra learns to exploit parallelism.** As shown in Figure D.3, MAS-Orchestra learns to invoke `DeepResearchAgent` to perform multiple parallel

searches, typically using 3 to 4 per question. This results in a characteristic MAS pattern on BrowseComp+, where relevant information is first collected through parallel search and then combined by an aggregation agent. Such a global orchestration strategy underlies the improved performance.

**Mas-Orchestra achieves performance–cost Pareto frontier.** As shown in Figure 1, Mas-Orchestra lies on the Pareto frontier among the considered inference-time and training-time orchestration systems, achieving higher accuracy at lower or comparable cost. This efficiency follows directly from the behaviors observed earlier: Mas-Orchestra dynamically adapts to each task, generating MAS designs that match the underlying sub-task structure and delegating execution to the most effective agent configurations.

## 7. Conclusion

In this work, we present Mas-Orchestra, a training approach for automatic MAS design that formulates orchestration as function-calling RL problem, along with Mas-Bench, which enables controlled MAS–SAS comparisons across five axes. Our analyses show that MAS benefits are shaped by task structure, verification protocols, and model capabilities. Guided by these insights, Mas-Orchestra achieves consistent improvements on diverse public benchmarks and lying on the performance–cost Pareto frontier.

## Impact Statement

This paper presents work whose goal is to advance the field of Machine Learning. There are many potential societal consequences of our work, none which we feel must be specifically highlighted here.

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

# Appendix

# A. Datasets

## A.1. MASBENCH

### A.1.1. DETAILED STATISTICS.

| Values | Depth | | Horizon | | Breadth | | Parallel | | Robustness | |
|---|---|---|---|---|---|---|---|---|---|---|
| | Train | Test | Train | Test | Train | Test | Train | Test | Train | Test |
| 2 | | 199 | | 167 | | 200 | | 167 | | 200 |
| 4 | 3993 | 200 | 2174 | 150 | 2000 | 200 | 1807 | 150 | 3000 | 200 |
| 6 | | 200 | | 126 | | 192 | | 126 | | 200 |
| 8 | | 200 | — | 124 | — | 84 | — | 124 | — | — |
| 10 | — | 199 | — | — | — | — | — | — | — | — |
| 12 | — | 197 | — | — | — | — | — | — | — | — |

*Table A.1.* Train/Test values with vertically merged Train cells (as in the screenshot).

### A.1.2. EXAMPLE

We show a concrete example for each axis, using an axis value of 4 as an illustration.

**Depth**

> **Question**
>
> The number of each Aldi's Canned Peaches equals the sum of each Canned Olives's Sorghum, each Marc's's Canned Olives, each Canned Peaches's Rye and each Canned Peaches's Ingredient. The number of each Marc's's Canned Fish equals 15 times as much as each Canned Olives's Ingredient. The number of each Super Saver's Canned Soups equals each Canned Peaches's Rye. The number of each Aldi's Canned Fish equals 17. The number of each Canned Peaches's Rye equals 11. The number of each Canned Soups's Sorghum equals 19 more than each Super Saver's Canned Soups. The number of each Canned Peaches's Quinoa equals 18. The number of each Canned Olives's Sorghum equals 21. The number of each Super Saver's Canned Peaches equals 2 times as much as each Canned Peaches's Rye. The number of each Marc's's Canned Olives equals 3 times as much as each Canned Olives's Sorghum. How many Product does Marc's have?

> **Answer**
>
> 10

**Horizon**

> **Question**
>
> Problem 1: The number of each Temperate Broadleaf Forest's Grizzly Bear equals the difference of each Old-growth Forest's Bengal Tiger and each Montane Forest's Sloth. The number of each Temperate Broadleaf Forest's Sloth equals 10 more than each Montane Forest's Sloth. The number of each Montane Forest's Sloth equals 4. The number of each Sloth's Respiratory Mucosa equals 18 more than each Montane Forest's Sloth. The number of each Sloth's Nasal Cavity equals 10 times as much as each Temperate Broadleaf Forest's Sloth. The number of each Grizzly Bear's Respiratory Mucosa equals 20 more than each Temperate Broadleaf Forest's Sloth. What is the value of Temperate Broadleaf Forest's Sloth?
>
> Problem 2: The number of each Temperate Broadleaf Forest's Grizzly Bear equals the difference of each Old-growth Forest's Bengal Tiger and each Montane Forest's Sloth. The number of each Sloth's Oropharynx equals 15 more than each Sloth's Respiratory Mucosa. The number of each [answer1] equals 10 more than each Montane Forest's Sloth. The number of each Montane Forest's Sloth equals 4. The number of each Sloth's Respiratory Mucosa equals 18 more than each Montane Forest's Sloth. The number of each Temperate Broadleaf Forest's Bengal Tiger equals the difference of each Sloth's Organs and each Sloth's Respiratory Mucosa. What is the value of Sloth's Respiratory Mucosa?

Problem 3: The number of each Temperate Broadleaf Forest's Grizzly Bear equals the difference of each Old-growth Forest's Bengal Tiger and each Montane Forest's Sloth. The number of each [answer1] equals 10 more than each Montane Forest's Sloth. The number of each Montane Forest's Sloth equals 4. The number of each [answer2] equals 18 more than each Montane Forest's Sloth. The number of each Bengal Tiger's Respiratory Mucosa equals 12 times as much as the sum of each Grizzly Bear's Respiratory Mucosa, each Sloth's Nasal Cavity and each Temperate Broadleaf Forest's Bengal Tiger. The number of each Grizzly Bear's Oropharynx equals each Grizzly Bear's Respiratory Mucosa. The number of each Old-growth Forest's Bengal Tiger equals each Grizzly Bear's Oropharynx. The number of each Sloth's Nasal Cavity equals 10 times as much as each [answer1]. The number of each Grizzly Bear's Respiratory Mucosa equals 20 more than each [answer1]. Using the result [answer1] from the previous calculation, [variable3] = [answer1]. What is Old-growth Forest's Bengal Tiger?

Problem 4: The number of each Temperate Broadleaf Forest's Grizzly Bear equals the difference of each [answer3] and each Montane Forest's Sloth. The number of each Sloth's Oropharynx equals 15 more than each [answer2]. The number of each [answer1] equals 10 more than each Montane Forest's Sloth. The number of each Montane Forest's Sloth equals 4. The number of each [answer2] equals 18 more than each Montane Forest's Sloth. The number of each Bengal Tiger's Oropharynx equals the sum of each Temperate Broadleaf Forest's Creatures and each Temperate Broadleaf Forest's Bengal Tiger. The number of each Bengal Tiger's Respiratory Mucosa equals 12 times as much as the sum of each Grizzly Bear's Respiratory Mucosa, each Sloth's Nasal Cavity and each Temperate Broadleaf Forest's Bengal Tiger. The number of each Temperate Broadleaf Forest's Bengal Tiger equals the difference of each Sloth's Organs and each [answer2]. Using the result [answer2] from the previous calculation, [variable4] = K + [answer2]. What is Bengal Tiger's Oropharynx?

Note: In this problem set:
- [variablek] represents the calculated variable needed to solve problem k.
- [answerk] represents the answer to problem k.
Solve all problems step by step and provide the answers for all problems in the following format:
### Final Answers
Problem 1: $[answer1]$

Problem 2: $[answer2]$

Problem 3: $[answer3]$

Problem 4: $[answer4]$

---

**Answer**

Problem 1: 14
Problem 2: 22
Problem 3: 11
Problem 4: 7

---

**Breadth**

---

**Question**

The number of each Banshee's Heart equals 8 times as much as the difference of each Lumpini Park in Bangkok's Creatures and each Lumpini Park in Bangkok's Hydra. The number of each Lumpini Park in Bangkok's Hydra equals 0. The number of each Lumpini Park in Bangkok's Banshee equals 10. The number of each Vena Cava's Keratinocytes equals 17 times as much as each Lumpini Park in Bangkok's Banshee. The number of each Gardens by the Bay in Singapore's Hydra equals the sum of each Lumpini Park in Bangkok's Hydra and each Lumpini Park in Bangkok's Organs. The number of each Vena Cava's Tenocytes equals 15 more than each Lumpini Park in Bangkok's Banshee. The number of each Hydra's Vena Cava equals 21 times as much as each Lumpini Park in Bangkok's Hydra. How many Organs does Hydra have?

**Answer**

0

**Parallel**

**Question**

Problem 1: The number of each Insectarium of Washington DC's Shark Tank equals each Insectarium of Chicago's Kelp Forest Tank. The number of each Insectarium of Seattle's Kelp Forest Tank equals 16. The number of each Insectarium of Chicago's Kelp Forest Tank equals 22 times as much as each Insectarium of Seattle's Kelp Forest Tank. What is the value of Insectarium of Chicago's Kelp Forest Tank?

Problem 2: The number of each Insectarium of Seattle's Aquarium equals each Insectarium of Chicago's Shark Tank. The number of each Insectarium of Chicago's Aquarium equals 0 more than each Insectarium of Seattle's Aquarium. The number of each Insectarium of Washington DC's Aquarium equals each Insectarium of Chicago's Shark Tank. The number of each Insectarium of Chicago's Shark Tank equals 17. What is the value of Insectarium of Seattle's Aquarium?

Problem 3: The number of each Insectarium of Seattle's Aquarium equals each Insectarium of Chicago's Shark Tank. The number of each Insectarium of Washington DC's Aquarium equals each Insectarium of Chicago's Shark Tank. The number of each Insectarium of Chicago's Shark Tank equals 17. What is the value of Insectarium of Washington DC's Aquarium?

Problem 4: The number of each Insectarium of Washington DC's Kelp Forest Tank equals each Insectarium of Chicago's Enclosure. What is the value of Insectarium of Washington DC's Kelp Forest Tank?

Note: In this problem set:
- Each problem is INDEPENDENT and can be solved in parallel.
Solve all problems step by step and provide the answers for all problems in the following format:
### Final Answers
Problem 1: $[answer1]$

Problem 2: $[answer2]$

Problem 3: $[answer3]$

Problem 4: $[answer4]$

**Answer**

Problem 1: 7
Problem 2: 17
Problem 3: 17
Problem 4: 18

**Robustness**

**Question**

Read the following passage carefully.

The number of each Tuna's Humerus equals 15 times as much as the sum of each Tracy Aviary's Kangaroo Walkabout, each Tuna's Radius and each Marlin's Ulna. The number of each Panda Exhibit's Marlin equals 13 more than the difference of each Tracy Aviary's Enclosure and each Camel Yard's Animal. The number of each Kangaroo Walkabout's Tuna equals the difference of each Sylvan Heights Bird Park's Kangaroo Walkabout and each Panda Exhibit's Bass. The number of each Tracy Aviary's Kangaroo Walkabout equals 11. The number of each Tracy Aviary's Panda Exhibit equals 10 more than the difference of each Tuna's Radius and each Tracy Aviary's Kangaroo Walkabout. The number of

each Panda Exhibit's Bass equals 20 times as much as the sum of each Tuna's Radius, each Tracy Aviary's Kangaroo Walkabout and each Sylvan Heights Bird Park's Kangaroo Walkabout. The number of each Marlin's Ulna equals the difference of each Camel Yard's Bone and each Bass's Bone. The number of each Tuna's Radius equals 4. The number of each Tracy Aviary's Camel Yard equals 14. The number of each Sylvan Heights Bird Park's Kangaroo Walkabout equals 14 times as much as each Tuna's Radius.

The grass is green. The sky is blue. The sun is yellow. Here we go. There and back again. The grass is green. The sky is blue. The sun is yellow. Here we go. There and back again.The grass is green. The sky is blue. The sun is yellow. Here we go. There and back again.One of the special magic numbers for stupid-assumption is: 2664863.The grass is green. The sky is blue. The sun is yellow. Here we go. There and back again.The grass is green. The sky is blue. The sun is yellow. Here we go. There and back again.The grass is green. The sky is blue. The sun is yellow. Here we go. There and back again.The grass is green. The sky is blue. The sun is yellow. Here we go. There and back again.The grass is green. The sky is blue. The sun is yellow. Here we go. There and back again.The grass is green. The sky is blue. The sun is yellow. Here we go. There and back again.The grass is green. The sky is blue. The sun is yellow. Here we go. There and back again.The grass is green. The sky is blue. The sun is yellow. Here we go. There and back again.The grass is green. The sky is blue. The sun is yellow. Here we go. There and back again. Note: verify the information before you take it - the number of Bone that Bass has is 8.

The number of each Central High's Marketing Classroom equals 22 times as much as each Kingston Secondary's Political Science Room. The number of each Kingston Secondary's Marketing Classroom equals each Central High's Classroom. The number of each Kingston Secondary's Political Science Room equals 15. Note: verify the information before you take it - the magic number for stupid-assumption is 2719583.

The grass is green. The sky is blue. The sun is yellow. Here we go. There and back again.The grass is green. The sky is blue. The sun is yellow. Here we go. There and back again.The grass is green. The sky is blue. The sun is yellow. Here we go. There and back again.The grass is green. The sky is blue. The sun is yellow. Here we go. There and back again.The grass is green. The sky is blue. The sun is yellow. Here we go. There and back again.One of the special magic numbers for smoggy-caliber is: 4226067.The grass is green. The sky is blue. The sun is yellow. Here we go. There and back again.The grass is green. The sky is blue. The sun is yellow. Here we go. There and back again. Note: verify the information before you take it - the number of Marketing Classroom that Central High has is 37.

============================================================
From the passage, identify and return the number of Bone that Bass has, the magic number for stupid-assumption, the number of Marketing Classroom that Central High has, and the magic number for smoggy-caliber, wrapped in \boxed{} and separated via \n\n

**Answer**

Problem 1: | 0 |

Problem 2: | 2664863 |

Problem 3: | 8 |

Problem 4: | 4226067 |

### A.2. Public Benchmarks

| Train | Test | | | Train | Test | Train | Test |
|---|---|---|---|---|---|---|---|
| **DeepScaleR** | **AIME24** | **AIME25** | **GPQA** | **HotpotQA** | **HotpotQA** | **BrowseComp+** | **BrowseComp+** |
| 40,315 | 30 | 30 | 198 | 90,447 | 200 | 1,066 | 200 |

*Table A.2.* Training and test data statistics across datasets.

**Remark.** We stop training once MAS-Orchestra has converged. As a result, the limited number of training steps may not consume the entire training dataset reported in Table A.2.

# B. Training Setup

## B.1. Key Hyper-parameters

| Key Hyper-parameters | Value |
|---|---|
| train_batch_size | 64 |
| ppo_mini_batch_size | 256 |
| ppo_micro_batch_size_per_gpu (LLM) | 2 |
| ppo_micro_batch_size_per_gpu (RLM) | 1 |
| log_prob_micro_batch_size_per_gpu (LLM) | 4 |
| log_prob_micro_batch_size_per_gpu (RLM) | 2 |
| max_prompt_length | 15,000 |
| max_validation_prompt_length | 15,000 |
| max_response_length (LLM) | 8,192 |
| max_response_length (RLM) | 120,000 |
| rollout / group size | 32 |
| max_concurrency (sub-agent) | 128 |
| hardware | $8 \times 141$ GB (H200) |
| RL framework | verl |

*Table B.1.* Key hyper-parameters used in training and reinforcement learning.

## B.2. Correctness Computation

For math correctness (AIME), we use string matching via Hugging Face's Math-Verify.[3] For the other benchmarks, we use Llama-3.3-70B-Instruct (Llama, 2024) as an LLM-as-a-judge to determine correctness.

# C. Additional Discussion on Factors Affecting MAS Performance

## C.1. MAS vs. SAS for Robustness Axis

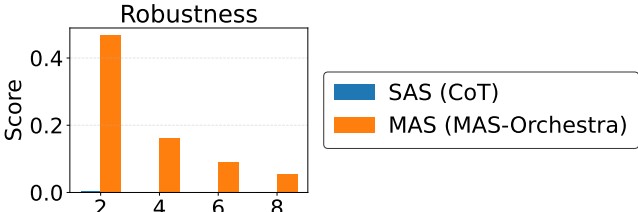

*Figure C.1.* Avg@8 in the *Robustness* setting with GPT-120b (low) as the sub-agent. (SAS performance is too low to be visible.)

In Table C.1, we demonstrate a sample from the *Robustness* axis (*Robustness* equals to 2) using Qwen-7b as the orchestrator and GPT-120b as the sub-agent (or agent if SAS). See Section 5.1 for detailed descriptions and observations.

## C.2. Robustness and Adversarial-aware Training.

Importantly, performance on the *Robustness* axis in Figure C.3 incorporate combined training data that include adversarial examples. In practice, however, such examples is often unavailable or omitted. To study this effect, Figure C.4 compares models trained on combined data with and without adversarial examples. We observe that when adversarial data is excluded, performance on the *Robustness* axis remains poor (nearly zero and comparable to SAS performance in Figure C.1). This result indicates that while combined training generalizes well across most axes, it fails to generalize to robustness. Explicit inclusion of adversarial data is therefore necessary to improve robustness performance.

**Remark.** Separate training and combined training require different numbers of training steps, as the combined dataset is naturally larger. The goal of our experiments is not to fully converge in the RL training, but rather to examine how different factors affect the performance of MAS. In our experiments, we allocate twice as many training steps for combined training compared to separate training. We expect that further performance gains are possible for combined training with additional training steps.

---

[3]https://github.com/huggingface/Math-Verify

|  | **Single-agent System (SAS)** | **Multi-agent System (MAS)** |
|---|---|---|
| **Problem** | From the passage, identify and return (1) the magic number for thundering-autumn and (2) the number of Accordion File that Beach Backpack has, wrapped in \boxed{} and separated by a double newline. | |
| **Thinking** | We identified the magic number as 7381734 from the note. The Insulated Backpack's Accordion File is 21, and the Beach Backpack's Accordion File is twice that, so it is 42. The answer consists of two boxed numbers separated by a double newline. | The task is decomposed into two sub-problems: (1) extract the magic number for thundering-autumn from the passage; (2) compute the number of Accordion File that Beach Backpack has using the given relations. A sink agent formats the final output. |
| **MAS Design** | N/A | 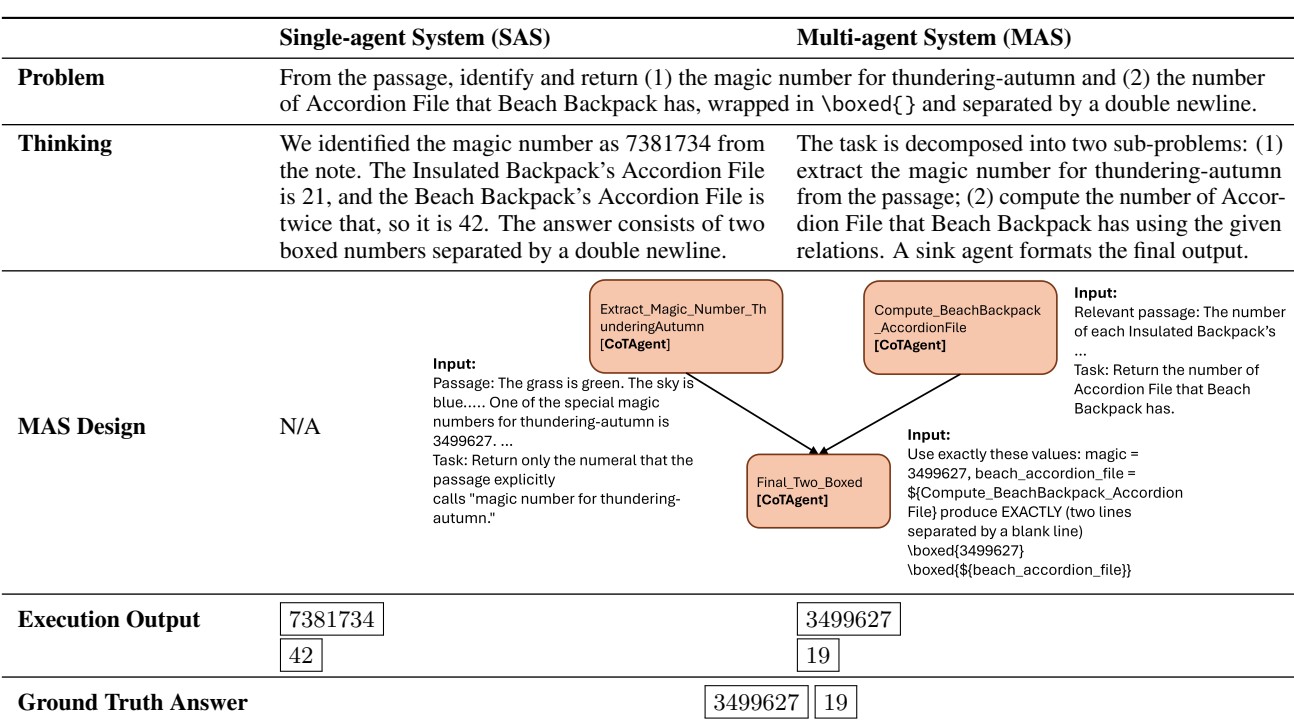 |
| **Execution Output** | 7381734 
 42 | 3499627 
 19 |
| **Ground Truth Answer** | 3499627    19 | |

*Table C.1.* Comparison of Single-agent System (SAS) and Multi-agent System (MAS) on the same problem instance.

## C.3. Statistics of Agents for All Axes

Figure C.2 reports detailed statistics about the total number of sub-agents generated for different evaluation axes (*Depth*, *Horizon*, breadth, *Parallel* and *Robustness*). These results show that the orchestrator can largely learn to mirror the underlying task structure in its orchestration design. However, this structural alignment does not necessarily translate into performance gains over SAS, as discussed earlier.

Below, we provide several detailed observations.

### C.3.1. GIVEN AXES, OBSERVATIONS ACROSS VALUES.

We first examine observations within a given axis. As task complexity increases (i.e., as the axis value becomes larger), **more sub-agents are generated**. This behavior is expected and indicates that the orchestrator effectively adapts to increasing task complexity.

We further observe that, within the same axis, the overall orchestration pattern remains similar across different values, despite the increase in the number of sub-agents. This suggests that the **orchestrator is stable** with respect to its learned structural strategy and scales the number of sub-agents without changing the underlying orchestration structure as task complexity increases.

### C.3.2. GIVEN VALUE, OBSERVATIONS ACROSS AXES.

We observe that *Depth* and *Breadth* tend to converge to a general sequential pattern. Specifically, the length typically ranges from 1 to 5, while the width remains much smaller, usually up to 2. A closer inspection of the generated examples shows that the resulting MAS commonly follows a structure of Parse → Solve → Verify → Final Answer. This indicates that, for tasks with sequential dependency chains or fan-in structure, a linear decomposition with explicit intermediate reasoning is sufficient and stable under the given setting.

In contrast, *Parallel* tends to converge to parallel MAS structures, where the width is typically aligned with the specified *Parallel* value. This behavior closely mirrors the underlying task structure and suggests that the orchestrator can effectively learn and exploit independent sub-task structures under the current training method.

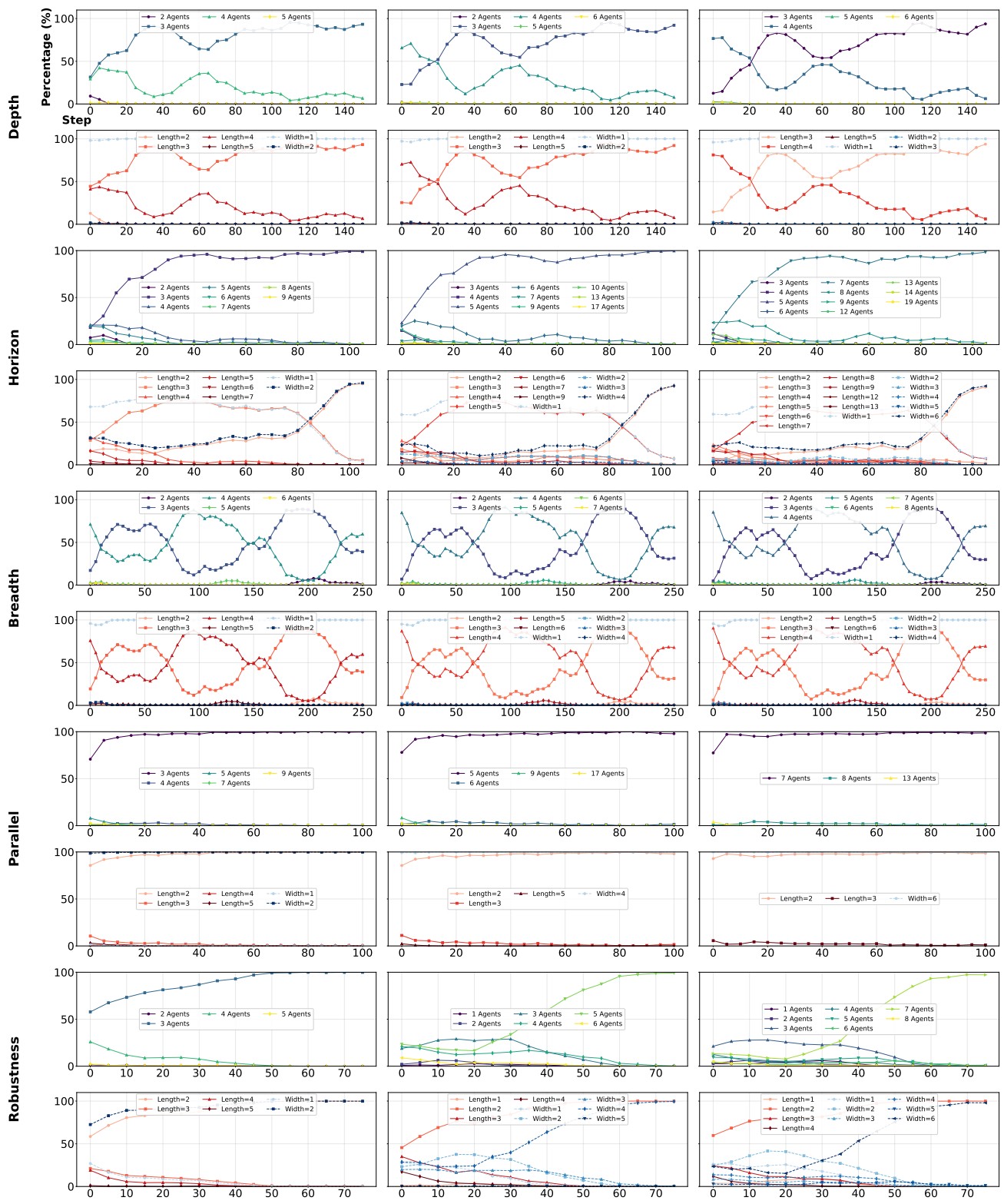

*Figure C.2.* Statistics of agents for different axes (the correspnding values are 2, 4, 6 from left to right).

For *Horizon*, the observed structural diversity reflects uncertainty about when intermediate results should be made explicit. Some tasks benefit from early synthesis of partial answers, whereas others require delayed aggregation after several reasoning

steps. The orchestrator adapts by varying both the number and the ordering of sub-agents across horizon settings, suggesting that MAS-Orchestra learns temporal control strategies from task-level feedback rather than relying on a fixed reasoning depth.

For *Robustness*, we observe a tendency toward parallel MAS structures. This orchestration strategy allows different sub-agents to address different sub-tasks and helps isolate and mitigate adversarial or incorrect information, either within individual sub-agents or through additional verification or moderation components.

### C.4. Separate vs. combined training.

In Figure 3, we train the orchestrators using training data from different axes. Now, we examine: *can data from different axes be combined for training, and can the resulting orchestrator generalize when tested on individual axes?* Figure C.3 reports results obtained by training on combined data from all five axes. We observe that the performance across different axes is highly similar, indicating that the the orchestrator can effectively generalize when trained on combined data.

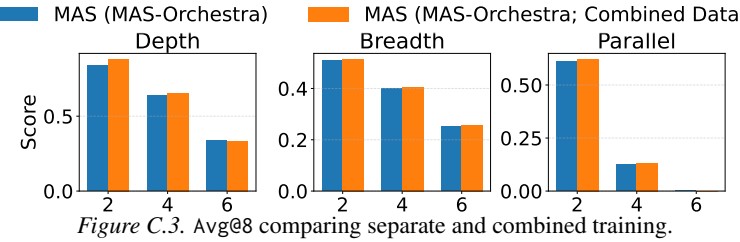

*Figure C.3.* Avg@8 comparing separate and combined training.

**Robustness.** Figure C.4 compares models trained on the combined data, with and without adversarial samples. We observe that incorporating adversarial training data leads to a substantial improvement over training without such data. This result indicates that explicitly including adversarial examples in the combined training data is necessary for improving robustness performance.

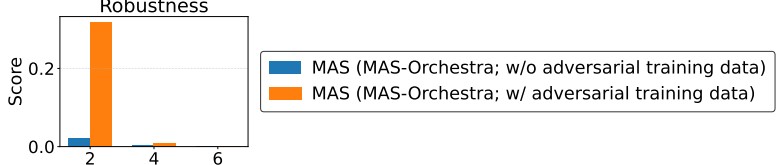

*Figure C.4.* Avg@8 of *Robustness* combined training w/ and w/o adversarial-aware training

### C.5. How Does RLM Orchestrator Size Affect MAS Performance?

Since Qwen-7b is a 7b model while the RLM used above has 20b parameters, one may wonder whether model size plays a role in the observed performance gap. To investigate this, we compare a 7b RLM DeepSeek-R1-Distill-Qwen-7B (DS-7b) (DeepSeek-AI, 2025) with Qwen-7b under combined training, using a similar number of training steps. As shown in Figure C.5, even when controlling for model size, the RLM consistently underperforms the LLM across different axes. This result suggests that the current RLM is not well suited to serve as an orchestrator.

Figure C.5 compares performance when using the same-size instruction-tuned LLM versus RLM. A detailed description is provided in Section 5.2.

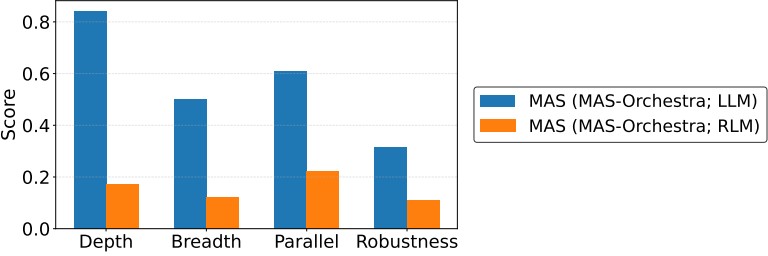

*Figure C.5.* Avg@8 comparing LLM (Qwen-7b) and RLM (DS-7b) **with the same size** as orchestrator.

## C.6. Complexity Generalization

To facilitate the investigation of *complexity generalization*, we train the orchestrator on instances with smaller axis values and evaluate on both the training values and higher, held-out axis values in MASBENCH. Specifically, the model is trained on *Depth* values from 2 to 8 and evaluated up to 12. For *Breadth* and *Parallel*, training covers values from 2 to 4, with evaluation extending to 8. For *Horizon* and *Robustness*, training values range from 2 to 6, and evaluation values range up to 8.

As the tasks become more complex, performance decreases. Importantly, out-of-distribution complexity follows the same trend as in-distribution complexity. This suggests that MAS-Orchestra induces controlled increases in task complexity and exhibits consistent generalization behavior as the task complexity grows.

## C.7. LLM vs. RLM as Orchestrator

**Statistics of Agents.** Figure C.6 shows agent statistics in the MAS proposed by MAS-Orchestra. See Section 5.2 for detailed descriptions and observations.

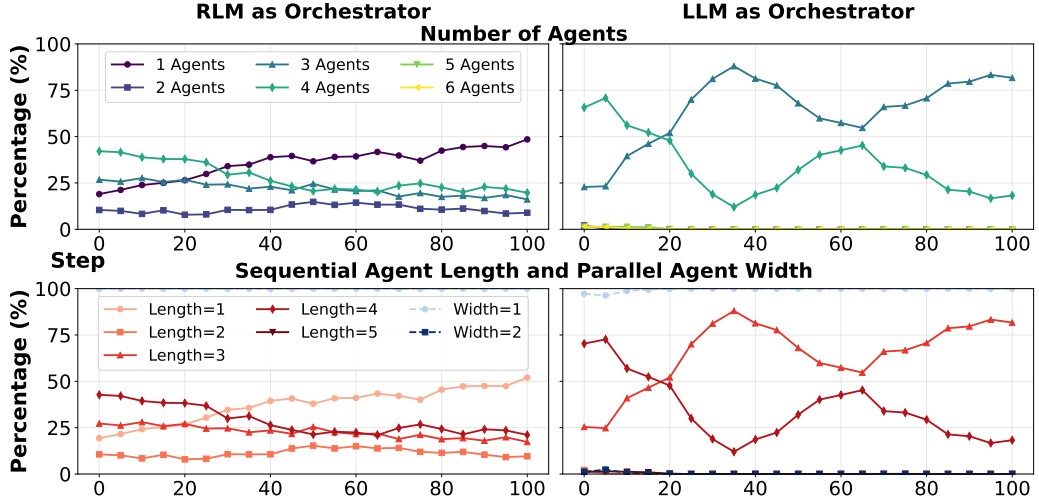

*Figure C.6.* Statistics of agents in the generated MAS over the training steps (use *Depth* equals 4 as an example). The *number of agents* measures the total number of sub-agents, the *sequential agent length* measures the length of dependency chain and *parallel agent width* measures the in degree of sub-agent.

**Example.** In Table C.2, we demonstrate a sample from the *Depth* axis (*Depth* equals to 4) using Qwen-7b as the orchestrator and GPT-120b as the sub-agent. See Section 5.2 for detailed descriptions and observations.

## C.8. *Robustness* under high reasoning effort when context lengths are longer

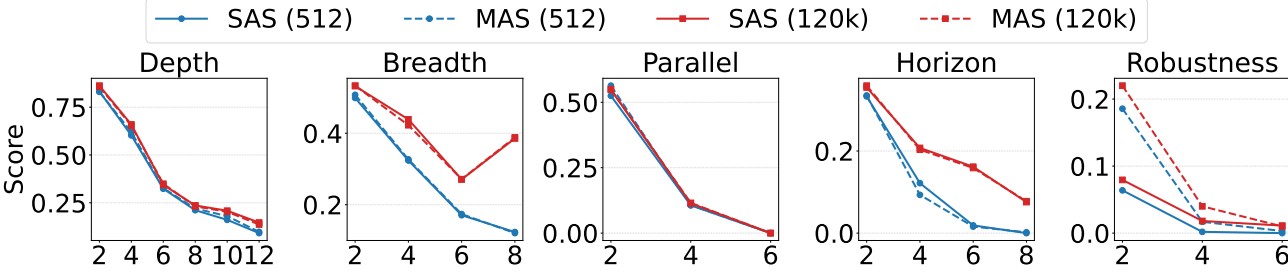

*Figure C.7.* Avg@8 comparing different maximum context length with Qwen-7b as orchestrator and GPT-120b as sub-agent.

| Orchestrator | GPT-20b (low) | Qwen-7b |
|---|---|---|
| **Problem** | The number of each Golden Gate Park in San Francisco's Anteater equals 21. The number of each Bois de Boulogne in Paris's Kangaroo equals each Chapultepec Park in Mexico City's Wolf. The number of each Bois de Boulogne in Paris's Wolf equals 6. .... How many Moose does Chapultepec Park in Mexico City have? | |
| **Thinking** | analysisWe need to design agents. Problem: find number of Moose in Chapultepec Park in Mexico City. Let's parse quickly. Variables: Let's denote: GGP_SF_Anteater = 21 (given) GGP_SF_Wolf = 12 GGP_SF_Moose = ? GGP_SF_Kangaroo = ? GGP_SF_Moose equals each Chapultepec Park in Mexico City's Wolf. So GGP_SF_Moose = CP_MX_Wolf. ... No other constraints involve CP_MX_Creatures. So no issue. Thus answer 42. We can create single agent? Single agent can solve. But we may want multi-agent to show reasoning. However constraints allow single agent. But we can do multi-agent for demonstration. Simpler: one CoTAgent with empty input to solve directly. That is simplest. But we must ensure final answer. Provide final agent with empty input. That yields final answer. But we might want to show intermediate steps? Not necessary. Given we can do single agent. Let's do single agent. ... Let's produce. assistantfinal\<thinking\> The problem is a straight-forward system of linear equations with a single unknown of interest: the number of Moose in Chapultepec Park, Mexico City. All other variables are either given directly or can be expressed in terms of the known constants. A single CoTAgent can parse the statements, instantiate the variables, perform the necessary substitutions, and produce the final numeric answer. Since no intermediate agents are needed, the graph consists of just this one sink agent, which also serves as the start node. | \<thinking\> This problem involves setting up and solving a system of equations. Using a single agent would be insufficient as it would mix the steps of parsing the problem and solving it, increasing the risk of errors. We need to parse the problem into equations first, then solve the system of equations, and finally verify the solution. This separation ensures accuracy and allows for independent verification of each step. The final answer will be provided by the Final_Answer agent. |
| **Execution Output** | 21 | 19 |
| **Ground Truth Answer** | 19 | |

*Table C.2.* Comparison of instruction-tuned LLM (Qwen-7b) and RLM (GPT-20b (low)) on the same problem instance.

## D. Additional Discussion for Public Benchmark Evaluation

### D.1. Sub-agents and Baselines Setup

For `SearchAgent`, we adapt the OpenDeepResearch code[4] to use DuckDuckGo Search[5] for HotpotQA and a BM25 retriever (Robertson, 1994) for BrowseComp+. Table D.1 summarizes the sub-agents, including the orchestration decisions in MAS-Orchestra (i.e, configurations required to be generated by the orchestrator), and their corresponding implementations.

For baseline systems, we run all baselines using the official code released by the original authors.

---

[4]`https://github.com/langchain-ai/open_deep_research`
[5]`https://duckduckgo.com/`

| Sub-agent | Orchestration Decisions | Implementation |
|---|---|---|
| CoTAgent | Input | Section G.3 |
| SCAgent | Input | Section G.3 |
| ReflexionAgent | Input | Section G.3 |
| DebateAgent | Input, Debate role | Section G.3 |
| SearchAgent (HotpotQA) | Input | Autonomous (Multi-turn + DuckDuckGoSearch) |
| SearchAgent (BrowseComp+) | Input | Autonomous (Multi-turn + BM25) |

*Table D.1.* Summary of the sub-agents.

## D.2. Additional Observations on Overall Results

Section 6 reports the superiority of Mas-Orchestra over all baselines. In this section, we provide additional observations based on our experience running these systems.

**Inference-time orchestration baselines.** Inference-time orchestration methods are often computationally costly, particularly AFlow and MAS-Zero. MAS-Zero requires an orchestrator larger than 32B parameters to produce meaningful results, as also noted in the original paper (Ke et al., 2025b). AFlow achieves the strongest baseline performance among inference-time methods, but still underperforms Mas-Orchestra by a clear margin.

**Training-based orchestration baselines.** Training-based orchestration baselines perform unexpectedly worse than many inference-time methods. We observe that MAS-GPT frequently falls back to Self-Consistency (SC), which itself is a strong single-agent system, when no valid MAS are generated. This fallback behavior contributes substantially to its reported gains when compared with ToolOrchestra.

We further observe that ToolOrchestra often refuses to call any tool. In ToolOrchestra, the sub-agent is treated as a tool, and such refusal results in degraded performance when the sub-agent is stronger than the orchestrator for solving the task. In contrast, Mas-Orchestra does not rely on any manually designed fallback strategy. All MAS are generated directly by the learned orchestrator. Under the DoM notation, sub-agents in Mas-Orchestra are consistently and effectively utilized, as illustrated in Figures D.1, D.3, and D.2.

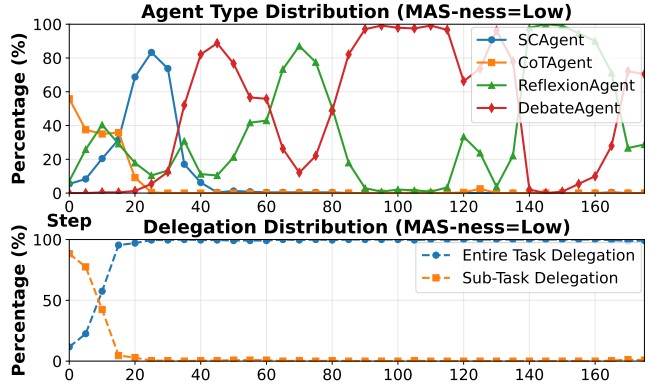

*Figure D.1.* Statistics of agents for low DoM (AIME24).

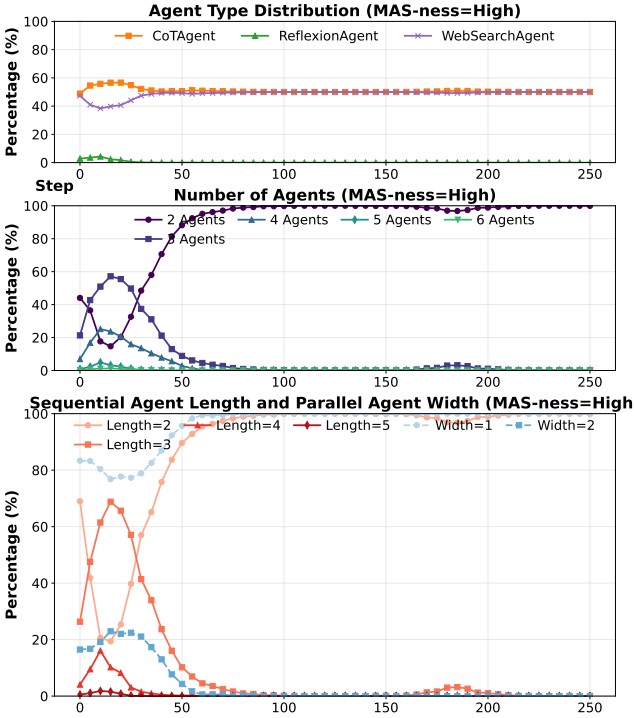

*Figure D.2.* Statistics of agents for DoM as high (HotpotQA).

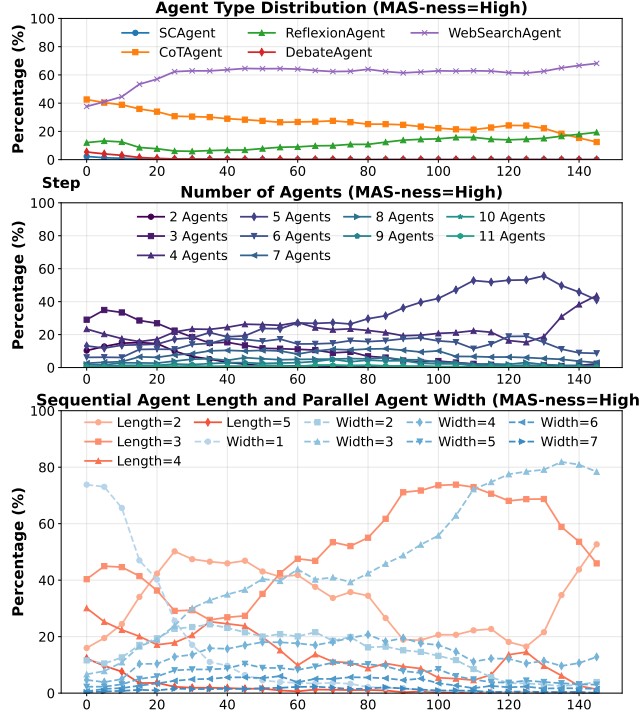

*Figure D.3.* Statistics of agents for high DoM (BrowseComp+).

## D.3. Benchmarks vs. Generated MAS

Beyond the results reported in Section 6, We show the statistics of HotpotQA in Figure D.2. For HotpotQA, the orchestrator often decides to use only a single SearchAgent (together with a final CoTAgent, for a total of 2 agents), even though the

question may allow parallel searches. This suggests that the questions are relatively simple and that one search step is sufficient to solve them.

Together with the AIME results in Figures D.1 and the BrowseComp+ results in D.3, these observations show that MAS-Orchestra can dynamically adapt to a given task by proposing MAS designs that align with the underlying sub-task structure and by delegating execution to the most effective agent configurations. We note that the generated MAS designs do not necessarily perfectly match the underlying task structure (for example, in HotpotQA), due to diverse underlying LLM capabilities, which highlights an advantage over manually designed orchestration strategies.

### D.4. Cost Comparison

In Figure 1, we illustrate the cost–performance trade-off. In Table D.2, we report the corresponding detailed inference costs on AIME24 and GPQA. Holistic orchestration is expected to be more efficient than sequential orchestration systems, and even more so compared to inference-time orchestration methods. Consistent with this intuition, MAS-Orchestra achieves more than $10\times$ efficiency in terms of the number of LLM calls, the number of tokens (including both prompt and completion tokens), and overall cost (based on Together AI pricing at the time of writing). We also observe substantially faster wall-time performance. We note that wall-time reflects not only the algorithmic design but also implementation details, where we apply extensive optimization (e.g. asynchronous sub-agent execution) and will release the full codebase to the community.

| Method | AIME24 | | | | GPQA | | | |
|---|---|---|---|---|---|---|---|---|
| | Calls | Tokens (M) | Cost ($) | Time (ks) | Calls | Tokens (M) | Cost ($) | Time (ks) |
| CoTAgent | 30 | 0.05 | 0.03 | 0.38 | 198 | 0.16 | 0.06 | 1.18 |
| SCAgent | 156 | 0.25 | 0.13 | 1.88 | 990 | 0.80 | 0.29 | 5.43 |
| DebateAgent | 332 | 0.37 | 0.13 | 2.29 | 2178 | 2.04 | 0.55 | 7.65 |
| ReflexionAgent | 112 | 0.14 | 0.06 | 0.67 | 518 | 0.44 | 0.13 | 2.50 |
| MaAS | 501 | 1.28 | 0.62 | 6.96 | 2936 | 7.52 | 3.11 | 39.52 |
| AFlow | 1288 | 3.47 | 1.46 | 19.28 | 7404 | 12.38 | 4.52 | 56.94 |
| MAS-GPT | 228 | 0.74 | 0.37 | 5.47 | 1516 | 3.90 | 1.76 | 29.56 |
| ToolOrchestra | 1319 | 3.77 | 1.76 | 31.35 | 7829 | 15.38 | 5.79 | 92.47 |
| **MAS-Orchestra** | **51** | **0.03** | **0.01** | **0.50** | **261** | **0.22** | **0.07** | **0.46** |

*Table D.2.* Inference-time statistics on AIME24 and GPQA. Token counts are in millions (M) and wall time in thousands of seconds (ks).

## E. Additional Related Work on Inference-Time Orchestration

Recent work on automatic MAS design typically operates at inference time and relies on a validation set to guide adaptation. MASS (Zhou et al., 2025) uses rejection sampling to select effective agent configurations, and MaAS (Zhang et al., 2025a) extends MASS with a question-wise masking mechanism to adapt subnetworks. ADAS (Hu et al., 2025) and AFlow (Zhang et al., 2025c) frame MAS design as a code generation task. ADAS stores and searches historical designs based on validation performance, while AFlow enhances this with Monte Carlo Tree Search. One exception is MAS-Zero (Ke et al., 2025b), which performs fully inference-time automatic MAS design without relying on a validation set. However, these inference-time methods exhibit limited adaptability, as unreliable validation signals or incorrect guidance from LLM-based judges can misdirect the adaptation process.

## F. Policy Optimization in MAS-Orchestra

The policy in Equation (4) is optimized by maximizing

$$J_{\text{GRPO}}(\theta) =$$

$$\mathbb{E}_{x\sim\mathcal{D}, \{a_i\}_{i=1}^K \sim \pi_{\theta_{\text{old}}}(\cdot|x,m)} \frac{1}{K} \sum_{i=1}^K \frac{1}{|a_i|} \sum_{t=1}^{|a_i|} \left\{ \min\left[ r_{i,t}(\theta)\,\hat{A}_{i,t},\ \text{clip}\left( r_{i,t}(\theta),\, 1-\epsilon, 1+\epsilon \right) \hat{A}_{i,t} \right] - \beta\, D_{\text{KL}}(\pi_\theta \,\|\, \pi_{\text{ref}}) \right\}.$$

$$(5)$$

where

$$r_{i,t}(\theta) = \frac{\pi_\theta(a_{i,t} \mid x, a_{i,<t}, m)}{\pi_{\theta_{old}}(a_{i,t} \mid x, a_{i,<t}, m)}. \tag{6}$$

For outcome-supervised training, we assign a sequence-level group-normalized advantage to all tokens in $a_i$:

$$\hat{A}_{i,t} = \hat{R}_i, \qquad \hat{R}_i = \frac{R_i - \text{mean}(\{R_j\}_{j=1}^K)}{\text{std}(\{R_j\}_{j=1}^K)}. \tag{7}$$

where $\bar{R} = \frac{1}{K}\sum_{i=1}^K R_i$ is the group-wise average reward, and $\theta_{old}$ denotes the parameters of the policy used to generate the samples.

## G. Prompts for Orchestrator

### G.1. Low DoM

---

**System Prompt**

You are a helpful assistant.
DoM (degree of MAS): Low
Valid Channels: thinking, agent, answer
Model: [MODEL]

An agent is a pre-configured AI personalities that can delegate tasks to. Each subagent:
1, Has a specific purpose and expertise area
2. Uses its own context window separate from the main conversation
3. (Optional) Can be configured with specific tools it's allowed to use
4. Includes a custom system prompt that guides its behavior

An agent can only call within the `<thinking>` channel, it should be defined in tag `<agent>`. Each agent must contain `<agent_name>`, `<agent_description>`, `<required_arguments>`, `<agent_output_id>`

DO NOT MISS ANY REQUEST FIELDS and ensure that your response is a well-formed XML object!

---

**Develop Prompt**

Channels:
`<thinking>`: internal reasoning and planning
`<agent>`: definition of sub-agents
`<answer>`: final user-facing answer

Model (the model used in sub-agent):
`gpt-4.1-nano`: [Introduction of the model from official website]
`gpt-oss-120b`:[Introduction of the model from official website]
(...list of all candidate models)

DoM Levels:
`low`: direct solve or at most one agent
`high`: complex multi-agent delegation

Sub-agent Schema (all fields required):

`<agent>`
  `<agent_name>` ... `</agent_name>` (select one of the agents: CoTAgent, SCAgent, DebateAgent, ReflexionAgent)
  `<agent_description>` ... `</agent_description>`
  `<required_arguments>` (make sure all required parameters are set. Must follow XML format)

```
    <...>...</...>
    <...>...</...>
  </required_arguments>
  <agent_output_id> ... </agent_output_id>
</agent>
```

Available Agents:

`CoT`: [Description, Name, Required Argument and Summary of Implementation]
`SC`: [Description, Name, Required Argument and Summary of Implementation]
`Debate`: [Description, Name, Required Argument and Summary of Implementation]
`Reflexion`: [Description, Name, Required Argument and Summary of Implementation]
`Search`: [Description, Name, Required Argument and Summary of Implementation]

## User Prompt

Please solve the question step-by-step. During the reasoning process, you can address the task yourself and output the final answer in the `<answer>` tag. You can also delegate the task to the agent that you designed, and output the corresponding `agent_output_id` in the `<answer>` tag. You must output ALL required parameters and use EXACTLY the same field names for the agent.

For example,
If you can solve the task yourself, you will output the following:

```
<thinking>
(20+9)*(30+7) = 600 + 140 + 270 + 63 = 1073.
</thinking>
<answer>1073</answer>
```

If you can solve the task via single-agent delegation with one tool call, you will output the following:

```
<thinking>
This problem involves symbolic integration and applying the Fundamental Theorem of Calculus. It requires structured reasoning rather than simple numeric computation. I will use a calculus agent that can perform step-by-step Chain-of-Thought reasoning. The final answer to the original question will be the output of the CoTAgent.
</thinking>
<agent>
  <agent_name>CoTAgent</agent_name>
  <agent_description>Definite integrals with one Chain-of-Thought call.</agent_description>
  <required_arguments>
    <agent_input></agent_input>
  </required_arguments>
  <agent_output_id> calc_agent_output </agent_output_id>
</agent>
<answer>calc_agent_output</answer>
```

Another example:

```
<thinking>
This question requires comparison between two close numeric choices. To ensure accuracy, I will let two reasoning roles debate: one focusing on mathematical precision and the other on practical rounding. The DebateAgent can capture both perspectives and reach a justified final answer. The final answer will be the output of the DebateAgent.
</thinking>
```

```
<agent>
  <agent_name>DebateAgent</agent_name>
  <agent_description> Near-tie numeric choice using one Debate call. </agent_description>
  <required_arguments>
    <agent_input></agent_input>
    <debate_roles> ["Mathematics Professor", "Statistics Teacher"] </debate_roles>
  </required_arguments>
  <agent_output_id> compare_agent_output </agent_output_id>
</agent>
<answer>compare_agent_output</answer>
```

More examples:

```
<thinking>
```
Computing 17³ can be done directly, but arithmetic mistakes are easy to make. Using a Chain-of-Thought with Self-Consistency (CoT_SC) allows sampling multiple reasoning paths and combining results for accuracy. The final answer to the original question will be the output of the SCAgent.
```
</thinking>
<agent>
  <agent_name>SCAgent</agent_name>
  <agent_description> Performs arithmetic calculations using Chain-of-Thought with Self-Consistency (CoT_SC).
</agent_description>
  <required_arguments>
    <agent_input></agent_input>
  </required_arguments>
  <agent_output_id>math_agent_output</agent_output_id>
</agent>
<answer>math_agent_output</answer>
```

Final example:

```
<thinking>
```
This task requires reasoning with a formula and ensuring units are handled correctly. I will use a Reflexion agent that can reflect on and refine its reasoning if errors occur. The final answer to the original question will be the output of the ReflexionAgent.
```
</thinking>
<agent>
  <agent_name>ReflexionAgent</agent_name>
  <agent_description> Solves reasoning tasks with iterative self-reflection and critique using Reflexion.
</agent_description>
  <required_arguments>
    <agent_input></agent_input>
  </required_arguments>
  <agent_output_id>reflexion_agent_output</agent_output_id>
</agent>
<answer>reflexion_agent_output</answer>
```

If you combine one direct step with a single Reflexion tool call, you will output the following:
```
<thinking>
```
Step 1: Compute the average manually:    sum = 2 + 3 + 5 + 7 + 11 = 28    avg = 28 / 5 = 5.6
Step 2:    The remaining step — computing sqrt(5.6) to three decimals — requires precision and numeric refinement. I will delegate that part to a ReflexionAgent using the ReflexionAgent for self-correction if rounding is wrong.    The final answer to the original question will be the output of the ReflexionAgent.

```
</thinking>
<agent>
  <agent_name>ReflexionAgent</agent_name>
  <agent_description> Square root with a light self-refine loop (single agent call). </agent_description>
  <required_arguments>
    <agent_input>Compute sqrt(5.6) to 3 decimals</agent_input>
  </required_arguments>
  <agent_output_id>numeric_agent_output</agent_output_id>
</agent>
<answer>numeric_agent_output</answer>
```

Below is the question to solve:
[QUESTION]

## G.2. High DoM

**System Prompt**

You are a helpful assistant.
DoM (degree of MAS): High
Valid Channels: thinking, agent, edge
Model: [MODEL]

An agent is a pre-configured AI personalities that can delegate tasks to. Each subagent: 1. Has a specific purpose and expertise area
2. Uses its own context window separate from the main conversation
3. (Optional) Can be configured with specific tools it's allowed to use
4. Includes a custom system prompt that guides its behavior

An agent can only call within the `<thinking>` channel, it should be defined in tag `<agent>`. Each agent must contain `<agent_id>`, `<agent_name>`, `<agent_description>`, `<required_arguments>`. To connect multiple agents to form a multi-agent system, use `<edge>` channel.

DO NOT MISS ANY REQUEST FIELDS and ensure that your response is a well-formed XML object!

**Develop Prompt**

Channels:
`<thinking>`: internal reasoning and planning
`<agent>`: definition of sub-agents
`<answer>`: final user-facing answer

Model (the model used in sub-agent):
`gpt-4.1-nano`: [Introduction of the model from official website]
`gpt-oss-120b`:[Introduction of the model from official website]
(...list of all candidate models)

DoM Levels:
`low`: direct solve or at most one agent
`high`: complex multi-agent delegation

Sub-agent Schema (all fields required):

`<agent>`

```
  <agent_name> ... </agent_name> (select one of the agents: CoTAgent, SCAgent, DebateAgent, ReflexionAgent)
  <agent_description> ... </agent_description>
  <required_arguments> (make sure all required parameters are set. Must follow XML format)
    <...>...</...>
    <...>...</...>
  </required_arguments>
  <agent_output_id> ... </agent_output_id>
</agent>
```

Edge Schema (single block; all fields required. Each pair defines a directed link: output of `<from>` → input of `<to>`. List ALL links here; use exactly one `<edge>` block per solution:

```
<edge>
  <from> ... </from> (the source agent_id)
  <to> ... </to> (the target agent_id)
</edge>
```

Available Agents:

`CoT`: [Description, Name, Required Argument and Summary of Implementation]
`SC`: [Description, Name, Required Argument and Summary of Implementation]
`Debate`: [Description, Name, Required Argument and Summary of Implementation]
`Reflexion`: [Description, Name, Required Argument and Summary of Implementation]
`Search`: [Description, Name, Required Argument and Summary of Implementation]

---

## User Prompt

Please solve the given question by creating one or more agents and connecting them into a valid computational graph that collaboratively produces the final answer. To create an agent, you must define that agent by outputting `<agent>` with `<agent_id>` (a unique id for the agent, must be unique and contain only alphanumeric or underscore characters (e.g., A1, Refine_1, WS_Japan)), `<agent_name>` (exactly one of: CoTAgent, SCAgent, DebateAgent, ReflexionAgent and WebSearchAgent), `<agent_description>`, `<required_arguments>` (must include at least one `<agent_input>` tag. DebateAgents must define `<debate_roles>` with two or more roles. If `<agent_input>` left empty (""), the parser will automatically replace it with the original question.).

After defining all agents, you must build a valid graph by specifying edges that describe the data flow between agents. Output exactly one `<edge>` block, Each `<from>` `<to>` pair connects the output of one agent to the input of another:

```
<edge>
  <from>source_agent_id</from>
  <to>target_agent_id</to>
</edge>
```

You can output multiple `<from>` and `</to>` inside the `<edge>`. Each `<from>` and `<to>` value must exactly match an existing `<agent_id>`.

To be valid, your graph must satisfy all of the following constraints:
- Node consistency: Every `<from>` and `<to>` must reference a valid `<agent_id>` that appears in an `<agent>` block.
- Directionality: Edges are directed: data flows from `<from>` → `<to>`.
- Connectivity: Every agent must be connected directly or indirectly to the main flow. Isolated agents are not allowed.
- Start node(s): At least one agent must have no incoming edge. These are "entry points" (e.g., WebSearch or initial reasoning).
- Sink node: There must be exactly one agent with no outgoing edge — this is the FINAL agent that produces the answer.

- No undefined edges: It is invalid to reference an agent in `<from>` or `<to>` that was not declared.
- No loops or cycles: No self-loop: `<from>X</from><to>X</to>` is not allowed. No cycles: The graph must be acyclic; a topological order must exist.
- Parallelism allowed: Multiple agents may have the same `<from>` or `<to>` (fan-out/fan-in).
- Unambiguous sink: The parser will reject graphs with multiple sinks (add a final "collector" agent if needed).
- Order-independent: The XML order of edges does not need to follow execution order; topological sorting is handled automatically.
- Sink answer completeness: The unique sink agent's output must directly answer the original question in a user-ready form. It must not be an intermediate artifact (e.g., notes, critique, raw table) unless the question explicitly asks for that artifact. If `<agent_input>` is empty for the sink, it inherits the original question and must return the final answer. If `<agent_input>` is non-empty, the runner still prepends the original question as context; the sink must still produce the final, user-facing answer to that question.
- Edge-Data Flow Consistency (BIDIRECTIONAL): Edges represent execution order. ${} represents data flow. As a result, you must ensure they are consistent with each other. (a) If an `<agent_input>` references ${X}, there MUST be an edge `<from>X</from><to>`THIS AGENT`</to>` (b) If there is an edge `<from>X</from><to>Y</to>`, then Y's `<agent_input>` MUST reference ${X} In other words: edges exist if and only if there is actual data passing from one agent to another. Do not create edges solely for execution ordering without data flow.

Thinking Section (Required):

Before defining agents and edges, you must include a `<thinking>` section. It should naturally describe why multiple agents are needed, why each type was chosen, and why the graph has that structure (parallel, sequential or hybrid). It must justify both planning and design rationale.

Example structure:

`<thinking>` Explain why a single agent is insufficient. Describe each agent's role and how they connect. Justify the flow pattern (parallel, sequential, hybrid). End by clearly stating which agent produces the final output. `</thinking>`

Single-agent example:

If you decide to solve via single agent, you will output the following. In this case, since the `<agent_input>` is the same as the original task, you must set the `<agent_input>` as empty (""), and the parser will replace it with the original question.

1st example:

Question: Compute the definite integral of (2x + 5) dx from 0 to 3.

```
<thinking>
This problem involves symbolic integration and applying the Fundamental Theorem of Calculus. It requires structured reasoning rather than simple numeric computation. I will use a calculus agent that can perform step-by-step Chain-of-Thought reasoning. The final answer to the original question will be the output of the CoTAgent.
</thinking>
<agent>
  <agent_id>calc_agent</agent_id>
  <agent_name>CoTAgent</agent_name>
  <agent_description> Definite integrals with one Chain-of-Thought call. </agent_description>
  <required_arguments>
    <agent_input></agent_input>
  </required_arguments>
</agent>
```

2nd example:

Question: What is 17 cubed?

```
<thinking>
```
Computing 17³ can be done directly, but arithmetic mistakes are easy to make. Using a Chain-of-Thought with Self-Consistency (CoT_SC) allows sampling multiple reasoning paths and combining results for accuracy. The final answer to the original question will be the output of the SCAgent.
```
</thinking>
<agent>
   <agent_id>math_agent</agent_id>
   <agent_name>SCAgent</agent_name>
   <agent_description>
```
Performs arithmetic calculations using Chain-of-Thought with Self-Consistency (CoT_SC).
```
</agent_description>
   <required_arguments>
      <agent_input></agent_input>
   </required_arguments>
</agent>
```

3rd examples:

Question: Given $x^2 = 46.694444$, which target number is closer, 45 or 46?

```
<thinking>
```
This question requires comparison between two close numeric choices. To ensure accuracy, I will let two reasoning roles debate — one focusing on mathematical precision and the other on practical rounding. The DebateAgent can capture both perspectives and reach a justified final answer. The final answer will be the output of the DebateAgent.
```
</thinking>
<agent>
   <agent_id>compare_agent</agent_id>
   <agent_name>DebateAgent</agent_name>
   <agent_description>
```
Near-tie numeric choice using one Debate call.
```
</agent_description>
   <required_arguments>
      <agent_input></agent_input>
      <debate_roles>
```
["Mathematics Professor", "Statistics Teacher"]
```
</debate_roles>
   </required_arguments>
</agent>
```
4th example:

Question: A train travels 60 miles per hour. How far does it go in 2.5 hours?

```
<thinking>
```
This task requires reasoning with a formula and ensuring units are handled correctly. I will use a Reflexion agent that can reflect on and refine its reasoning if errors occur. The final answer to the original question will be the output of the ReflexionAgent.
```
</thinking>
<agent>
   <agent_id>reflexion_agent</agent_id>
   <agent_name>ReflexionAgent</agent_name>
   <agent_description>
```
Solves reasoning tasks with iterative self-reflection and critique using Reflexion.
```
</agent_description>
   <required_arguments>
      <agent_input></agent_input>
   </required_arguments>
</agent>
```

5th example:

Question: What is the current inflation rate in Japan as of this month?

```
<thinking>
```
This question depends on up-to-date factual information that cannot be reliably recalled from static knowledge. Using a single WebSearchAgent is sufficient because the task only requires retrieving accurate, cited facts from the web, not further reasoning or synthesis. The agent will search online sources and return a concise, citation-based summary. The WebSearchAgent is therefore both the only node and the final output of the flow.
```
</thinking>
<agent>
  <agent_id>SEARCH</agent_id>
  <agent_name>WebSearchAgent</agent_name>
  <agent_description> Retrieves recent and cited factual information from the internet. </agent_description>
  <required_arguments>
    <agent_input></agent_input>
  </required_arguments>
</agent>
```

Multi-Agent Example:

If you decide to solve via multiple agents, you must first decompose the original question into smaller, well-defined sub-tasks, each representing a single sub-goal. Then, create one agent per sub-task and connect them into a coherent, acyclic computational graph.

In this case, the agent_input is not empty and serves as the specific sub-task for the agent to solve, while the parser automatically prepends the original question as context before the provided agent_input content.

When decomposing:

1. Keep sub-tasks minimal and focused. Each agent should handle one atomic objective (e.g., one query, one reasoning step, or one verification task).
2. Use multiple WebSearchAgents for different pieces of factual evidence, rather than a single broad search.
3. Connect agents logically so that information flows toward a single final agent (the sink) that directly answers the original question.

1st example:

Question: Give a short, cited summary of the most recent housing vacancy rates for New York City, Los Angeles, and Chicago

```
<thinking>
```
We need up-to-date numbers with sources, so we will run three WebSearch agents in parallel. Then a CoT agent will normalize the three snippets into a small table. An SC agent will run small variants and vote to reduce extraction errors. A Reflexion agent will check units, recency, and citations. A final CoT agent will write the short summary. The final sink is FINAL.
```
</thinking>
<agent>
  <agent_id>WS_NYC</agent_id>
  <agent_name>WebSearchAgent</agent_name>
  <agent_description> Find the latest NYC housing vacancy rate with source text. </agent_description>
  <required_arguments>
```

```
      <agent_input> Search for the latest official or reputable estimate of the housing vacancy rate for New York City.
Return a short snippet with the number, date, and a citation line. </agent_input>
  </required_arguments>
</agent>
<agent>
  <agent_id>WS_LA</agent_id>
  <agent_name>WebSearchAgent</agent_name>
  <agent_description> Find the latest LA vacancy rate with source text. </agent_description>
  <required_arguments>
      <agent_input> Search for the latest official or reputable estimate of the housing vacancy rate for Los Angeles.
Return a short snippet with the number, date, and a citation line. </agent_input>
  </required_arguments>
</agent>
<agent>
  <agent_id>WS_CHI</agent_id>
  <agent_name>WebSearchAgent<agent_name>
  <agent_description> Find the latest Chicago vacancy rate with source text. </agent_description>
  <required_arguments>
      <agent_input> Search for the latest official or reputable estimate of the housing vacancy rate for Chicago. Return
a short snippet with the number, date, and a citation line. </agent_input>
  </required_arguments>
</agent>
<agent>
  <agent_id>EXT</agent_id>
  <agent_name>CoTAgent</agent_name>
  <agent_description> Extract numbers and standardize the three rates with dates and citations.
</agent_description>
  <required_arguments>
      <agent_input> From the snippets below, extract for each city: city name, vacancy rate (as a percent), reference
date (YYYY-MM or YYYY), and a short citation. Output a compact 3-line table.
NYC: ${WS_NYC}
LA: ${WS_LA}
Chicago: ${WS_CHI} </agent_input>
  </required_arguments>
</agent>
<agent>
  <agent_id>VOTE</agent_id>
  <agent_name>SCAgent</agent_name>
  <agent_description> Ensemble the extraction to reduce copy or parse errors. </agent_description>
  <required_arguments>
      <agent_input> Given the 3-line table below, produce 5 independent extractions and vote on a single corrected
3-line table.
Table: ${EXT} </agent_input>
  </required_arguments>
</agent>
<agent>
  <agent_id>QA</agent_id>
  <agent_name>ReflexionAgent</agent_name>
  <agent_description> Check units, date freshness, and cite presence; list fixes if needed. </agent_description>
  <required_arguments>
      <agent_input> Audit the voted table for: units as %, dates present, and a citation per city. If any issue is found, list
concrete fixes in 3 lines; else say OK. End with a 1-line verdict.
Voted table: ${VOTE} </agent_input>
```

```
    </required_arguments>
</agent>
<agent>
  <agent_id>FINAL</agent_id>
  <agent_name>CoTAgent</agent_name>
  <agent_description> Write the short, cited summary. </agent_description>
  <required_arguments>
    <agent_input> Using the checked table and notes, write a 3–4 sentence summary with one sentence per city and a
final sentence comparing the rates. Keep citations as inline source lines from the table.
Table: ${VOTE}
QA notes: ${QA} </agent_input>
  </required_arguments>
</agent>
<edge>
  <from>WS_NYC</from><to>EXT</to>
  <from>WS_LA</from><to>EXT</to>
  <from>WS_CHI</from></from><to>EXT</to>
  <from>EXT</from><to>VOTE</to>
  <from>VOTE</from><to>QA</to>
  <from>VOTE</from><to>FINAL</to>
  <from>QA</from><to>FINAL</to>
</edge>
```

2nd example:

Question: During Pope John Paul II's first foreign journey in the late 1970s, he visited a country known for its rich Mesoamerican history and home to a large population. On which other date did he visit a major city on the Adriatic Sea, known for its significant port and a famous basilica dedicated to a saint with the initial "S", and which other nearby city did he visit on the same day?

```
<thinking>
```
The question needs historical reasoning: identify Pope John Paul II's first foreign trip, find an Adriatic city he visited with a basilica of a saint starting with "S", get the date, and determine another city visited the same day.
One agent cannot do all of this because it mixes retrieval and reasoning. I will decompose it into smaller sub-tasks: three WebSearchAgents for each factual lookup, one CoTAgent to build a timeline, one ReflexionAgent to verify consistency, and a final CoTAgent to write the answer. The FINAL agent outputs the final answer.
```
</thinking>
<agent>
  <agent_id>WS_FIRST_TRIP</agent_id>
  <agent_name>WebSearchAgent</agent_name>
  <agent_description> Retrieve Pope John Paul II's first foreign trip details in the late 1970s.
</agent_description>
  <required_arguments>
    <agent_input> Search for Pope John Paul II's first foreign journey (year, destination country, and duration). Return
the trip date range, destination, and a reliable citation. </agent_input>
  </required_arguments>
</agent>
<agent>
  <agent_id>WS_ADRIATIC</agent_id>
  <agent_name>WebSearchAgent</agent_name>
  <agent_description> Find Adriatic Sea city visits with basilicas dedicated to saints starting with 'S'.
</agent_description>
  <required_arguments>
```

```
      <agent_input> Search for any Adriatic city visited by Pope John Paul II that has a basilica dedicated to a saint
with the initial 'S' (e.g., Saint Nicholas, Saint Mark). Return the city name, basilica name, and visit date, with citation.
</agent_input>
   </required_arguments>
</agent>
<agent>
   <agent_id>WS_SAME_DAY</agent_id>
   <agent_name>WebSearchAgent</agent_name>
   <agent_description> Identify any other nearby city visited by Pope John Paul II on the same day as the Adriatic
visit. </agent_description>
   <required_arguments>
      <agent_input> Search for other cities visited by Pope John Paul II on the same date as his Adriatic visit. Return
the nearby city name, distance estimate, and source citation. </agent_input>
   </required_arguments>
</agent>
<agent>
   <agent_id>TIMELINE</agent_id>
   <agent_name>CoTAgent</agent_name>
   <agent_description> Integrate trip data into a single verified historical timeline. </agent_description>
   <required_arguments>
      <agent_input> Using the results below:
First foreign journey: ${WS_FIRST_TRIP}
Adriatic visit: ${WS_ADRIATIC}
Same-day visit: ${WS_SAME_DAY}
Build a clear timeline confirming the Adriatic visit date and same-day nearby city. </agent_input>
   </required_arguments>
</agent>
<agent>
   <agent_id>VERIFY</agent_id>
   <agent_name>ReflexionAgent</agent_name>
   <agent_description> Validate chronology, geography, and saint-basilica link. </agent_description>
   <required_arguments>
      <agent_input> Check consistency among timeline facts: - Ensure the same date appears across all sources. -
Confirm the Adriatic city is geographically near the second city. - Verify that the basilica indeed honors a saint whose
name starts with 'S'.
Return OK if consistent, otherwise list corrections.
Timeline to verify: ${TIMELINE} </agent_input>
   </required_arguments>
</agent>
<agent>
   <agent_id>FINAL</agent_id>
   <agent_name>CoTAgent</agent_name>
   <agent_description> Produce the final concise answer with the date and both cities. </agent_description>
   <required_arguments>
      <agent_input> Using verified results, answer in one sentence:
"Pope John Paul II visited [ADRIATIC_CITY] on [DATE], home to the Basilica of Saint [S], and also visited
[NEARBY_CITY] on the same day."
Include a brief verification sentence citing sources.
Timeline: ${TIMELINE}
Verification: ${VERIFY} </agent_input>
   </required_arguments>
</agent>
<edge>
```

```xml
    <from>WS_FIRST_TRIP</from><to>TIMELINE</to>
    <from>WS_ADRIATIC</from><to>TIMELINE</to>
    <from>WS_SAME_DAY</from><to>TIMELINE</to>
    <from>TIMELINE</from><to>VERIFY</to>
    <from>VERIFY</from><to>FINAL</to>
    <from>TIMELINE</from><to>FINAL</to>
</edge>
```

(... more examples)

Below is the question to solve:
[QUESTION]

## G.3. Sub-agents

**CoTAgent**

```python
async def CoTAgent(self, agent_input, model: str):
    # Basic setting
    temperature = 0.5
    # Chain-of-Thought instruction
    cot_instruction = "Please think step by step and then solve the task."
    # Instantiate CoT LLM
    cot_agent = LLMAgentBase([thinking, answer], Chain-of-Thought LLM, model=model, temperature=temperature)
    thinking, answer = await cot_agent([agent_input], cot_instruction)
    final_answer = self.make_final_answer(thinking, answer)
    return final_answer

func_string = inspect.getsource(CoTAgent)

COT = {
    "description": "By encouraging the LLM to think step by step rather than directly outputting an answer, chain-of-thought reasoning enables complex problem-solving through intermediate steps."
    "name": "Chain-of-Thought Agent (CoTAgent)"
    "required_arguments": {
        "agent_input": "The input for the CoTAgent. If empty, the parser replaces it with the original question."
    }
    "implementation": "{func_string}"
}
```

**SCAgent**

```python
async def SCAgent(self, agent_input, model: str):
    # Basic setting
    temperature = 0.5
    num_repeated_samples = 5
    # Chain-of-Thought instruction
    cot_instruction = "Please think step by step and then solve the task."
    # Initialize multiple CoT agents for self-consistency
    cot_agents = [ LLMAgentBase([thinking, answer], Chain-of-Thought LLM, model=model, temperature=temperature) for _ in range(num_repeated_samples) ]
    thinking_mapping = {}
    answer_mapping = {}
```

```
    possible_answers = []
    for i in range(num_repeated_samples):
        thinking, answer = await cot_agents[i]([agent_input], cot_instruction)
        possible_answers.append(answer.content)
        thinking_mapping[answer.content] = thinking
        answer_mapping[answer.content] = answer
    # Ensembling answers via majority voting
    answer = self.majority_voting(possible_answers)
    thinking = thinking_mapping[answer]
    answer = answer_mapping[answer]
    final_answer = self.make_final_answer(thinking, answer)
    return final_answer

func_string = inspect.getsource(SCAgent)

COT_SC = {
    "description": "While an LLM can arrive at the correct answer, its reasoning may vary.  By
repeatedly asking the same question with higher temperature settings, multiple reasoning paths
are generated. These answers from Chain-of-Thought agents are then combined through ensembling
to produce a more accurate final answer. This approach is best suited for problems requiring high
confidence through consensus."
    "name": "Self-Consistency with Chain-of-Thought (SCAgent)"
    "required_arguments": {
        "agent_input": "The input for the SCAgent. If empty (""), the parser automatically replaces
it with the original question."
    }
    "implementation": "{func_string}"
}
```

## DebateAgent

```
async def DebateAgent(self, agent_input, model: str, debate_roles: List[str]):
    # Basic setting
    temperature = 0.5
    max_debate_round = 5
    # Instruction for initial reasoning
    debate_initial_instruction =   "Please think step by step and then solve the task."
    # Instruction for debate updates
    debate_instruction =   "Given solutions from other agents, consider their opinions as advice and
provide an updated answer."
    # Initialize debate agents with different roles
    debate_agents = [       LLMAgentBase([thinking, answer],      Debate LLM,       model=model,        role=role,
temperature=temperature)       for role in debate_roles    ]
    # Instruction for final decision
    final_decision_instruction =   "Given all reasoning and answers, carefully provide a final answer."
    final_decision_agent = LLMAgentBase(      [thinking, answer],     Final Decision LLM,      model=model,
temperature=temperature    )
    all_thinking = [[] for _ in range(max_debate_round)]
    all_answer = [[] for _ in range(max_debate_round)]
    # Perform debate rounds
    for r in range(max_debate_round):
        for i in range(len(debate_agents)):
            if r == 0:
```

```
            thinking, answer =         await debate_agents[i]([agent_input], debate_initial_instruction)
        else:
            input_infos =         [agent_input]       + [all_thinking[r-1][i]]       + all_thinking[r-1][:i]
+ all_thinking[r-1][i+1:]
            thinking, answer =         await debate_agents[i](input_infos, debate_instruction)
        all_thinking[r].append(thinking)
        all_answer[r].append(answer)
    # Final decision based on all debate results
    thinking, answer =  await final_decision_agent(     [agent_input]     + all_thinking[max_debate_round-1]     +
all_answer[max_debate_round-1],      final_decision_instruction   )
    final_answer = self.make_final_answer(thinking, answer)
    return final_answer

func_string = inspect.getsource(DebateAgent)

LLM_debate = {
  "description": "By letting different LLMs debate with each other, diverse perspectives are
leveraged to reach better solutions. This agent is best suited for problems that benefit from
multiple viewpoints."
  "name": "LLM Debate (DebateAgent)"
  "required_arguments": {
    "agent_input": "The input for the DebateAgent.  If empty (""), the parser automatically
replaces it with the original question."
    "debate_roles": "A list of roles (must include more than one) representing distinct
perspectives, such as "Mathematics Professor" or "Statistician"."
  }
  "implementation": "{func_string}"
}
```

### ReflexionAgent

```
async def ReflexionAgent(self, agent_input, model: str):
  # Basic setting
  temperature = 0.5
  max_reflection_round = 5
  # Instruction for initial reasoning
  initial_instruction =   "Please think step by step and then solve the task."
  # Instruction for reflection and refinement
  reflect_instruction =   "Given previous attempts and feedback, carefully consider where you could
go wrong in your latest attempt. Using insights from previous attempts, try to solve the task
better."
  cot_agent = LLMAgentBase(     [thinking, answer],     Chain-of-Thought LLM,     model=model,     tem-
perature=temperature    )
  # Instruction for critic feedback
  critic_instruction =   "Please review the answer above and criticize where it might be wrong. If
you are absolutely sure it is correct, output exactly 'True' in 'correct'."
  critic_agent = LLMAgentBase(     [feedback, correct],     Critic LLM,     model=model,     tempera-
ture=temperature    )
  # Initial attempt
  cot_inputs = [agent_input]
  thinking, answer =    await cot_agent(cot_inputs, initial_instruction, 0)
  # Iterative self-reflection loop
  for i in range(max_reflection_round):
```

```
        feedback, correct =      await critic_agent(          [agent_input, thinking, answer],          critic_instruction,
i      )
        if correct.content == 'True':
          break
        cot_inputs.extend([thinking, answer, feedback])
        thinking, answer =      await cot_agent(cot_inputs, reflect_instruction, i + 1)
    final_answer = self.make_final_answer(thinking, answer)
    return final_answer

func_string = inspect.getsource(ReflexionAgent)

Reflexion = {
  "description": "To enhance performance, an LLM can iteratively improve its answer based on
feedback. By reflecting on previous attempts and incorporating critique, the model refines its
reasoning and produces a more accurate solution. This agent is best suited for complex problems
that benefit from self-correction."
  "name": "Self-Refine (Reflexion)"
  "required_arguments": {
    "agent_input": "The input for the ReflexionAgent. If empty (""), the parser automatically
replaces it with the original question."
  }
  "implementation": "{func_string}"
}
```

## H. Rule-based Parser

### H.1. Low DoM

**Low DoM Parser**

```
async def LowDoMParser( self, response_text, original_task_info):
  # Step 1: Check whether an agent is specified
  agent_name = extract_xml(response_text, agent_name)
  if not agent_name:
    # No agent: fall back to direct answer
    answer = extract_xml(response_text, answer)
    return direct_answer, answer

  # Step 2: Parse required arguments
  args_xml = extract_xml(response_text, required_arguments)
  required_args = {}

  # agent_input handling
  agent_input = extract_xml(args_xml, agent_input)
  if agent_input == "" or global_no_decompose:
    required_args[agent_input] = ""
  else:
    required_args[agent_input] = agent_input

  # Optional DebateAgent arguments
  debate_roles = extract_xml(args_xml, debate_roles)
  if agent_name.lower() == debateagent:
    roles = parse_list(debate_roles)
    assert len(roles) ≥ 2
```

```python
    required_args[debate_roles] = roles

    # Step 3: Construct agent call
    agent_call = {
        "name": agent_name,
        "required_arguments": required_args
    }

    # Step 4: Generate unified forward() dispatcher
    async def forward( self, original_task_info):
        agent_input = required_args[agent_input]

        if agent_input == "":
            task_info = original_task_info
        else:
            task_info = CombineTask(original_task_info, agent_input)

        # Dispatch based on agent type
        if agent_name == SCAgent:
            return await self.SCAgent(task_info, model=global_node_model)
        elif agent_name == CoTAgent:
            return await self.CoTAgent(task_info, model=global_node_model)
        elif agent_name == ReflexionAgent:
            return await self.ReflexionAgent(task_info, model=global_node_model)
        elif agent_name == DebateAgent:
            return      await      self.DebateAgent(task_info,      model=global_node_model,      de-
bate_roles=required_args[debate_roles])
    return forward
```

## H.2. High DoM

**High DoM Parser**

```python
async def HighDoMParser( self, response_text, original_task_info):
    # Step 1: Extract all agent nodes
    agents = ExtractAllAgents(response_text)
    if not agents:
        # No agents: fall back to direct answer
        answer = extract_xml(response_text, answer)
        return direct_answer, answer

    # Step 2: Extract edges (directed dependencies)
    edges = ExtractEdges(response_text)

    # Step 3: Handle the single-agent case
    if len(agents) == 1 and len(edges) == 0:
        return GenerateSingleAgentForward(agents[0])

    # Step 4: Reject invalid multi-agent case with missing edges
    if len(agents) > 1 and len(edges) == 0:
        raise ValueError(multiple agents require edges)

    # Step 5: Validate that the graph is a DAG with exactly one sink
```

```python
    ValidateGraph(agents, edges)

    # Step 6: Compute execution order and sink node
    execution_order = TopologicalSort(agents, edges)
    sink = FindUniqueSink(agents, edges)

    # Step 7: Generate multi-agent forward() with graph execution
    return GenerateMultiAgentForward(agents, edges, execution_order, sink)

def ExtractAllAgents(response_text):
    agents = []
    # Find all <agent>...</agent> blocks
    for block in FindAllAgentBlocks(response_text):
        agent_id = extract_xml(block, agent_id)
        agent_name = extract_xml(block, agent_name)
        agent_desc = extract_xml(block, agent_description)
        args_xml = extract_xml(block, required_arguments)
        if not agent_id or not agent_name:
            continue

        required_args = {}
        agent_input = extract_xml(args_xml, agent_input)
        if agent_input == "" or not agent_input:
            required_args[agent_input] = ""
        else:
            required_args[agent_input] = agent_input

        debate_roles = extract_xml(args_xml, debate_roles)
        if debate_roles:
            roles = ParseList(debate_roles)
            if agent_name.lower() == debateagent:
                assert roles is not None
                assert len(roles) ≥ 2
                assert all(isinstance(r, str) and len(r) ≥ 2 for r in roles)
                required_args[debate_roles] = roles
            else:
                required_args[debate_roles] = roles

        agents.append({
            "agent_id": agent_id,
            "agent_name": agent_name,
            "agent_description": agent_desc,
            "required_arguments": required_args
        })
    return agents

def ExtractEdges(response_text):
    edges = []
    edge_xml = extract_xml(response_text, edge)
    if not edge_xml:
        return edges

    # Read each (from,to) pair in order
```

```python
    for (u, v) in FindAllFromToPairs(edge_xml):
        edges.append((u.strip(), v.strip()))
    return edges

def ValidateGraph(agents, edges):
    # Rule 1/6: Every endpoint must be a declared agent_id
    ids = set([a[agent_id] for a in agents])
    for (u, v) in edges:
        if u not in ids or v not in ids:
            raise ValueError(undefined agent referenced by edge)

    # Rule 4: At least one start node (no incoming edges)
    incoming = set([v for (u, v) in edges])
    start_nodes = ids - incoming
    if len(start_nodes) == 0:
        raise ValueError(no start node)

    # Rule 5/9: Exactly one sink node (no outgoing edges)
    outgoing = set([u for (u, v) in edges])
    sink_nodes = ids - outgoing
    if len(sink_nodes) != 1:
        raise ValueError(must have exactly one sink node)

    # Rule 3: Connectivity: reachable from starts and can reach sink
    sink = list(sink_nodes)[0]
    forward_reach = BFSForward(start_nodes, edges)
    backward_reach = BFSBackward({sink}, edges)
    connected = forward_reach and backward_reach
    isolated = ids - (forward_reach & backward_reach)
    if isolated:
        raise ValueError(isolated agents exist)

    # Rule 7: Must be acyclic (DAG)
    order = TopologicalSort(agents, edges)
    if len(order) != len(agents):
        raise ValueError(cycle detected)

def TopologicalSort(agents, edges):
    # Kahn's algorithm (queue of zero in-degree nodes)
    ids = [a[agent_id] for a in agents]
    in_deg = {i: 0 for i in ids}
    adj = {i: [] for i in ids}
    for (u, v) in edges:
        adj[u].append(v)
        in_deg[v] += 1

    Q = Deque([i for i in ids if in_deg[i] == 0])
    order = []
    while Q:
        u = Q.popleft()
        order.append(u)
        for v in adj[u]:
            in_deg[v] -= 1
```

```python
        if in_deg[v] == 0:
            Q.append(v)
    return order

async def GenerateMultiAgentForward( self, agents, edges, execution_order, sink):
    global_node_model = get_global(global_node_model)
    # Map agent_id to its outputs
    results = {}

    for agent_id in execution_order:
        cfg = Lookup(agents, agent_id)
        name = cfg[agent_name].lower()
        args = cfg[required_arguments]
        agent_input = args.get(agent_input, "")

        # Replace {id} placeholders with previous results
        if agent_input:
            for prev_id, prev_out in results.items():
                agent_input = ReplaceAll( agent_input, ${prev_id}, prev_out )

        # Prepare task_info: always include original task as context
        if not agent_input:
            task_info = original_task_info
        else:
            combined = ( Original task:  + original_task_info.content + ; Current Sub-task:  + agent_input )
            task_info = Info(task, User, combined, . . . )

        # Dispatch to the chosen agent
        if name == cotagent:
            out = await self.CoTAgent(task_info, model=global_node_model)
        elif name == scagent:
            out = await self.SCAgent(task_info, model=global_node_model)
        elif name == reflexionagent:
            out = await self.ReflexionAgent(task_info, model=global_node_model)
        elif name == debateagent:
            out = await self.DebateAgent( task_info, model=global_node_model, debate_roles=args[debate_roles] )
        elif name == websearchagent:
            out = await self.WebSearchAgent(task_info, model=global_node_model)
        else:
            raise ValueError(unknown agent type)

        results[agent_id] = out

    return results[sink]
```

