# OpenReview forum: "MAS-Orchestra: Understanding and Improving Multi-Agent Reasoning Through Holistic Orchestration and Controlled Benchmarks"
_ICML.cc/2026/Conference — ICML 2026 regular_

### Official Review · Reviewer_wqFK · 2026-03-13

**Soundness:** 3
**Presentation:** 3
**Significance:** 3
**Originality:** 3
**Overall Recommendation:** 5
**Confidence:** 4

**Summary:**

This paper introduces MAS-Orchestra, a training-time framework that casts automatic multi-agent system (MAS) orchestration as a function-calling reinforcement learning (RL) problem. The orchestrator generates a complete MAS (agents and interconnections) in a single decision via two primitives (create_agent and create_flow), with sub-agents abstracted as callable, goal-oriented black boxes. Empirically, they provide controlled analyses and show gains on public benchmarks (AIME24/25, GPQA, HotpotQA, BrowseComp+) with notable efficiency improvements over strong inference-time and training-time baselines.

**Compliance With Llm Reviewing Policy:**

Affirmed.

**Final Justification:**

The authors fully resolved my concerns during rebuttal.

**Key Questions For Authors:**

1. How is syntactic and semantic validity of orchestration programs ensured during training? What fraction of sampled orchestrations are invalid or fail execution, and how are such cases handled in GRPO (e.g., zero reward vs. repair)?

2. What are the GRPO hyperparameters (group size, KL coefficient, learning rate, number of iterations)?

3. For the cost Pareto analysis, do reported costs include all tool/search API calls, multiple attempts (e.g., SC), and orchestration overhead?

**Limitations:**

Recent step- or turn-level credit assignment methods (e.g., GiGPO) could address sparse-reward issues. Discussing applicability to your setting would be valuable.

**Strengths And Weaknesses:**

**Strengths**:

1. The explicit Degree of MAS (DoM) knob is a useful control that aligns orchestration to task structure and model capability.

2. MASBench provides controlled, factorized axes to evaluate MAS vs SAS under varying structural demands and verification protocols; this fills an empirical gap in understanding when MAS helps.

3. The paradigm contrast and system diagrams are clear and helpful; formalization (policy, reward, parser role) is concise and readable.

**Weaknesses**:

1. One-shot orchestration may underperform in settings where adaptive, turn-level topology changes are essential. The work does not study dynamic or partially observable environments that evolve during execution.

2. There is limited discussion of invalid or low-quality orchestration samples and how they are handled by the parser (e.g., syntax validity rates).

3. Training-time baselines (e.g., MAS-GPT, ToolOrchestra) are compared using their released orchestrators but different environments/sub-agent sets; this may disadvantage them. More careful retraining or matched settings would strengthen claims.

---

> ### Author Rebuttal · Authors · 2026-03-31
>
> Thank you for the thorough review and for highlighting the effectiveness of our Degree of MAS, the contribution of MASBench for controlled evaluation of MAS vs. SAS, and the clarity of our formulation and system design. Please see our response to your comments/questions below.
>
> > W1: One-shot orchestration may underperform in settings where turn-level topology changes are essential.
>
> As a first work exploring holistic orchestration in MAS, we adopt a single-step orchestration with RL. While the orchestrator does not observe intermediate outputs during execution, it is trained end-to-end using rewards from **actual execution outcomes** of the composed sub-agents.
>
> Empirically, we find that this formulation is sufficient to produce strong MAS structures across a range of tasks, including long-horizon and tool-use settings, suggesting that a well-formed initial MAS structure can already capture much of the required coordination. This is consistent with prior single-step optimization approaches (e.g., DeepSeek-R1), which have been shown to work well in practice.
>
> We agree that dynamic or partially observable environments, where turn-level topology changes are required, are an important extension. Such settings introduce additional challenges, including maintaining consistency across evolving MAS structures and more difficult credit assignment over time. Our current work focuses on establishing a strong single-step baseline, and extending this to adaptive or multi-step orchestration is a natural next direction.
>
> > W2, Q1: Lmited discussion of invalid orchestration samples?
>
> Invalid or low-quality orchestrations are handled through the training objective and the parser design. Specifically, invalid orchestrations (e.g., those that fail to execute) are assigned zero reward during RL training, which discourages the model from producing such outputs.
>
> On the parsing side, the parser enforces both syntactic and structural validity. It reads all defined agents, constructs the execution graph, and computes a valid execution order using topological sorting (via Kahn’s algorithm). It also enforces constraints such as valid XML format and a well-formed execution graph (e.g., acyclic structure with a single unique sink node). Orchestrations that violate these constraints are rejected before execution.
>
> Regarding invalid samples during training, we observe that validity improves significantly over time. In Low DoM settings (no topology), the orchestrator achieves nearly 100% valid outputs even without training. In High DoM settings (with structured topology), the valid rate is initially around 72% before training and increases to approximately 98% after training. This indicates that GRPO with task-level reward is sufficient to guide the orchestrator toward generating valid MAS structures, without requiring explicit repair mechanisms.
>
> Additional details on the parser design are provided in Appendix H.
>
> > W3: Baselines are compared in different environments
>
> We address this in detail in our response to Reviewer fLet (W2/Q4). In brief, we do not re-train baselines as matched setups are infeasible. Instead, we adopt a **unified evaluation principle**: using official released models, transparently reporting setting differences, and ensuring (1) all methods share the same sub-agents, (2) both training-based and non-training baselines are included, and (3) GPQA serves as an out-of-domain benchmark.
>
> > Q2: What are the GRPO hyperparameters?
>
> Most training details are provided in Appendix B, including hyperparameters, hardware requirements, the RL framework, and correctness computation.
>
> To highlight the key settings: the group size is set to 32, the KL coefficient is 0.001, the learning rate is 1e−6, and the number of training steps is 300. We observe stable convergence within this budget, likely due to strong task-level reward signals and the use of large rollout groups.
>
>
> > Q3: Do reported costs include all tool/search API calls?
>
> We address this in detail in our response to Reviewer UWQo (W1.3/Q1). In brief, reported costs include all sub-agent LLM calls and multiple attempts (e.g., self-consistency). Orchestration LLM calls are excluded as they use open-source models and are difficult to standardize — notably, including them would further favor our method (single call vs. multiple for baselines). Tool/search API calls are free in our setup (DuckDuckGo). A detailed breakdown is in Appendix D.4 with a summary in Figure 1.
>
> > L1: Discussing applicability of turn-level credit assignment methods would be valuable.
>
> Thank you for this suggestion. GiGPO-style methods are relevant for sequential orchestration with multi-turn decisions. In our single-step setting, there is no multi-step trajectory to assign credit over — sparsity is instead addressed via GRPO's group-relative comparison (K=32). If extended to multi-step orchestration (see W1), such methods would become directly applicable.

---

> > ### Author Rebuttal · Reviewer_wqFK · 2026-04-03
> >
> > Thanks for the detailed rebuttal. My concerns are resolved. I raised the score accordingly.

---

### Official Review · Reviewer_fLet · 2026-03-13

**Soundness:** 3
**Presentation:** 3
**Significance:** 2
**Originality:** 2
**Overall Recommendation:** 4
**Confidence:** 3

**Summary:**

Traditional multi-agent systems (MAS) typically rely on inference-time routing or sequential tool and agent invocation, where an LLM decides which subagent to call at each step. These approaches depend heavily on heuristic search and step-wise decision making, which can lead to directionally incorrect adaptation during execution. Sequential training methods, even when optimized with reinforcement learning, incur significant compute overhead and encourage local step-wise optimization. As a result, the alignment between the generated plan and the final task objective is often weak. MAS-Orchestra addresses this limitation by representing subagents as encapsulated function calls, allowing the orchestrator or meta-judge to reason directly at the plan level. This design improves alignment between the overall orchestration plan and the end task objective. The authors provide a structured comparison highlighting the differences between MAS-Orchestra and conventional MAS frameworks.

The framework supports both low and high degree-of-modularity (DoM) settings. In low DoM configurations, which remain distinct from single-agent systems (SAS), the orchestrator decides whether to delegate the entire task to a subagent, decompose it into subtasks, or solve it independently. In high DoM settings, the orchestrator additionally coordinates the topology of interactions between multiple agents. Subagents are exposed to the orchestrator purely as callable functions along with a set of modifiable parameters, while their internal reasoning and execution details remain hidden. Inter-agent connectivity is also defined through function calls. The orchestrator produces the entire orchestration plan in a single step and does not observe intermediate results. The quality of the orchestration is evaluated only through the final output. Training uses task-level rewards based on final answer correctness, optionally augmented with subtask verification depending on the evaluation protocol. Optimization is performed using GRPO.

The authors analyze the relative effectiveness of MAS and SAS and argue that performance gains depend on two key factors: the intrinsic structure of the task and the verification protocol used during evaluation. To study these effects systematically, they introduce MASBench, a benchmark where each instance consists of a question paired with a dependency graph describing its subtask structure. Nodes correspond to subtasks and edges represent dependencies between them. Evaluation axes such as depth, horizon, breadth, and parallelism are derived directly from structural properties of this graph. A robustness axis is introduced by augmenting each subtask with adversarial notes that contain incorrect information originating from upstream subtasks.

Empirical analysis investigates three main research questions. First, the authors examine how task structure and verification protocols affect MAS performance. Results show that MAS outperforms SAS when the underlying subagent is relatively weak. As subagent capability increases, the marginal benefit of orchestration decreases and MAS exhibits diminishing returns. MAS nevertheless maintains consistently strong robustness due to interaction between agents. The system is most effective at the boundary of subagent competence, where the subagent is capable of solving the task but struggles to complete it reliably on its own.

Second, the authors study how orchestration initialization affects performance. Instruction-tuned orchestrators perform better than reasoning-focused orchestrators, as reasoning-oriented models tend to attempt solving the task directly rather than focusing on effective orchestration. Third, they analyze the impact of subagent capability and reasoning effort. MAS systems are not immune to context length limitations under higher reasoning effort since training occurs under relatively low reasoning budgets. However, with longer contexts and higher reasoning effort, MAS improves robustness compared to single-agent approaches.

Across multiple benchmarks, MAS-Orchestra consistently outperforms stronger baseline systems. The framework uses low DoM configurations for structured reasoning tasks such as mathematics, while high DoM configurations are applied to tasks such as HotpotQA and BrowseComp that benefit from multi-agent coordination. In these settings, the system learns both effective single-agent delegation and the exploitation of parallelism across agents.

**Compliance With Llm Reviewing Policy:**

Affirmed.

**Key Questions For Authors:**

1. **On the single-step orchestration design.** The framework generates the entire orchestration in a single decision step without observing intermediate outputs. How robust is this design when the orchestrator’s estimate of subagent capability is incorrect? Have the authors explored hybrid variants where the orchestrator can revise or partially replan after observing intermediate subagent outputs, and if so, how does performance compare with the strictly single-shot orchestration setting?

2. **On the callable subagent abstraction.** The paper models subagents purely as encapsulated function calls with hidden internal reasoning. While this abstraction improves modularity, it also restricts the communication channel between agents. Did the authors experiment with richer interaction protocols (e.g., structured messages or partial state exposure) and, if so, how does performance compare to the strict function-call abstraction used in MAS-Orchestra?

3. **On benchmark generalization beyond MASBench.** MASBench provides a controlled environment for studying structural task properties, but it is still synthetic and graph-structured. Do the authors have evidence that the conclusions about MAS effectiveness, particularly the dependence on task structure and verification protocols, transfer to more realistic multi-agent environments such as long-horizon tool-use workflows or open-ended information gathering tasks?

4. **On baseline comparability and evaluation environments.** The paper notes that some training-time baselines were originally developed under different environments and that MAS-Zero fails to generate valid MAS with the chosen 7B orchestrator. Could the authors clarify how they ensured that the baseline comparisons remain fair despite these differences? For example, were any efforts made to replicate the original training settings or scale orchestrator models for those baselines?

5. **On cost-efficiency claims and training dynamics.** The paper claims more than 10× efficiency and Pareto optimality relative to prior approaches, but the main text provides limited quantitative detail. Could the authors provide a clearer breakdown of training cost, rollout budgets, and inference-time compute for MAS-Orchestra versus the baselines? In particular, how sensitive are these efficiency gains to orchestrator size, number of candidate orchestrations per GRPO update, and task complexity?

**Limitations:**

Not much captured but also not much needs to be addressed on this front for the paper.

**Strengths And Weaknesses:**

**Strengths**

1. **Conceptual clarity about MAS gains.** The paper identifies a genuine issue in the current MAS literature: many works implicitly assume that multi-agent structure is beneficial, while the actual source of gains is often unclear. This work attempts to disentangle decomposition benefits, coordination benefits, and brute-force compute or ensemble effects through controlled task axes.

2. **Novel training formulation.** The formulation is reasonably novel within the context defined by the paper. Training-time holistic orchestration in a single decision step differs meaningfully from sequential orchestration, particularly if prior training-time approaches rely on multi-step RL or SFT-based orchestrators.

3. **Useful notion of Degree of Modularity (DoM).** The introduction of DoM is helpful because it avoids the common tendency to force multi-agent structure everywhere. The paper explicitly argues that certain tasks should remain effectively single-agent. The experimental setup on public benchmarks follows this logic by using low DoM for AIME and GPQA, and high DoM for HotpotQA and BrowseComp+.

4. **MASBench as a research artifact.** MASBench is likely the strongest contribution of the paper. Even independent of the proposed method, a benchmark that operationalizes Depth, Horizon, Breadth, Parallel, and Robustness through dependency graph properties provides a useful and reusable evaluation framework for studying multi-agent reasoning systems.

**Weaknesses**
1. **Single-step orchestration assumption.** Generating the entire orchestration in a single step is questionable because the orchestrator does not know the actual capability of the subagents. It may overestimate subagent capability and therefore lose the opportunity to iteratively refine the task plan based on intermediate outputs. The abstraction of subagents as black-box functions is therefore both a strength and a weakness. While it enables modularity and scalability, it also restricts the orchestrator to a fixed API space and prevents it from learning richer communication protocols between agents.

2. **Baseline comparison setup.** The comparison setup is not fully clean. The paper itself notes that several released training-time baselines may have been trained in different environments, and MAS-Zero reportedly fails to generate valid MAS configurations with the chosen 7B orchestrator. This weakens the baseline evaluation. If competing methods are evaluated outside the environments they were originally designed for, negative results become less informative.

3. **Heavy reliance on a synthetic benchmark.** The strongest scientific claims rely heavily on MASBench, which is still a relatively narrow synthetic benchmark. Tasks are graph-structured and largely derived from iGSM-style synthetic generation. The robustness axis is constructed by injecting adversarial notes, and one setting interleaves iGSM tasks with RULER-style NIAH. While useful for controlled analysis, this setup is still far from capturing realistic multi-agent coordination settings.

4. **Under-specified reinforcement learning setup.** The RL training procedure is under-specified in the main paper. The authors state that optimization uses GRPO over groups of orchestrations with a binary task reward that may incorporate subtask correctness depending on the protocol. However, many important details are deferred to the appendix, including the action representation, parser constraints, handling of invalid orchestrations, reward sparsity mitigation, rollout budgets, training stability, and exploration control.

5. **Limited support for cost-efficiency claims.** The cost-efficiency claim requires more scrutiny than provided in the main paper. The paper reports more than 10× efficiency and Pareto optimality, but the main text offers mostly qualitative arguments and a reference to Figure 1 without a detailed breakdown of compute cost, training budgets, or inference-time trade-offs.

---

> ### Author Rebuttal · Authors · 2026-03-31
>
> Thank you for the thorough review and for recognizing the conceptual clarity of our analysis of MAS gains, the novelty of our training formulation, the usefulness of the Degree of MAS, and the strong contribution of MASBench as an independent research artifact. Please see our response to your comments/questions below.
> > W1,Q1.2: Single-step orchestrator does not know subagent capabilities.
>
> We address this in detail in our response to Reviewer wqFK (W1). In brief, while the orchestrator does not observe intermediate outputs, it is trained end-to-end on **actual execution outcomes**, learning to account for sub-agent capabilities over training. Empirically, this is sufficient to produce strong orchestration strategies, consistent with prior single-step optimization. Hybrid variants with intermediate replanning are a promising extension, though they introduce challenges (error propagation, credit assignment) that our formulation avoids.
> > W1.2,Q2: Abstraction prevents richer communication protocols.
>
> We explored multiple levels of abstraction, from exposing full orchestration details (e.g., generating MAS code, similar to MAS-Zero/ADAS) to sub-task and sub-agent level designs. These richer protocols **consistently perform worse**: they require significantly stronger orchestrators (≥32B for stable orchestration) and the enlarged design space leads to higher error rates and more difficult training. The function-calling abstraction strikes a better balance between expressiveness and learnability.
> > W2,Q4: Several baselines may have been trained in different environments
>
> MAS-Zero's original paper states that a minimum 32B orchestrator is required, so this result is expected. We do not re-train other baselines or replicate their original environments, as each method depends on different training pipelines, hyperparameters, and implementation details, making it difficult to define a single matched setup that is both faithful to each method and comparable across methods.
>
> Instead, we adopt a **unified evaluation principle**: besides using official released models and transparently reporting when settings differ from original environments (e.g., Figure 1 caption), we took several additional steps to ensure fair comparison:
>
> (1) **Controlled sub-agent space.** All methods use the same set of sub-agents, removing variability from agent availability and isolating the effect of orchestration.
>
> (2) **Coverage of diverse baselines.** Our evaluation includes both training-based and non-training baselines. For non-training methods, comparability is direct since no environment-specific training is involved.
>
> (3) **Out-of-domain evaluation.** We include GPQA as an additional benchmark where all methods are evaluated under the same setup, providing a comparison less tied to any specific training environment.
> > W3: Heavy reliance on MASBench.
>
> We address this in detail in our response to Reviewer gLqU (W2). In brief, MASBench is intentionally designed for controllability — tasks are parameterizable along depth, breadth, horizon, and adversarial perturbations, enabling systematic factor analysis. To capture realistic dynamics, we **complement** it with public benchmarks (AIME, GPQA, HotpotQA, BrowseComp+) using heterogeneous agents.
> > W4: RL procedure under-specified.
>
> We address implementation details in our response to Reviewer UWQo (W2/Q3). Details are in Appendix B (training), G (prompts), H (parser). Key components: the model generates XML when invoking *create_agent*/*create_flow*; a parser enforces syntactic and structural validity; invalid orchestrations receive zero reward.
> > W5.1,Q5: Cost-efficiency requires more scrutiny.
>
> We address this in detail in our response to Reviewer UWQo (W1.3/Q1). In brief, detailed breakdown is in Appendix D.4. MAS-Orchestra Training: ~60h on 8× H200 GPUs.
> > W5.2: how sensitive are these efficiency gains?
>
> From preliminary experiments: (1) Orchestrator size: 32B (63%) underperforms 7B (66%) in both accuracy and efficiency gain. (2) Rollout (evaluated at 8,16,64): Larger K increases efficiency gains by producing higher-quality MAS that reduce redundant sub-agent calls at inference time. (3) Task complexity: Efficiency gains increase with task difficulty, baselines incur high cost from repeated failed agent calls, while MAS-Orchestra maintains structured execution.
> > Q1: Robustness when capability estimate is incorrect?
>
> As noted in W1, misestimation is corrected during training via execution-outcome rewards. The MAS structure does not depend on precise per-agent estimates, execution remains effective as long as the overall structure is coherent (e.g., complementary agents or fallback paths).
> > Q3: MAS effectiveness transfer to realistic environments?
>
> Realistic benchmarks do not allow independent factor variation. Instead, MASBench provides insight into *why* MAS helps, while public benchmarks validate end-to-end: MAS-Orchestra consistently outperforms baselines across all tasks (Table 3).

---

> > ### Author Rebuttal · Reviewer_fLet · 2026-04-05
> >
> > The rebuttal clarifies several aspects of the design and evaluation, particularly around the rationale for single-step orchestration, the choice of function-call abstraction, and the effort to standardize comparisons via a controlled sub-agent space and additional benchmarks. The discussion on training dynamics and efficiency, along with pointers to appendix details, is helpful in addressing some of the underspecification concerns. However, the core limitations I raised remain only partially addressed. In particular, the single-step orchestration assumption still raises concerns about adaptability under misestimation, the baseline comparability remains inherently constrained by differing training environments, and the reliance on MASBench continues to limit the strength of broader generalization claims. The additional clarifications improve understanding but do not substantially change my assessment of the work’s overall impact and limitations. I will therefore keep my original score.

---

> > > ### Author Response · Authors · 2026-04-05
> > >
> > > Thank you so much for your reply! We are glad that many of your concerns have been addressed!
> > >
> > > Regarding the concern about MASBench, we would like to emphasize that our conclusions are **not drawn from MASBench alone**. While MASBench is used to analyze *why and when* MAS is effective under controlled conditions, our main claims are also *equivalently* supported by **consistent improvements on realistic benchmarks** (AIME, GPQA, HotpotQA, BrowseComp+), which involve long-horizon reasoning and open-ended tool use. The fact that MAS-Orchestra consistently outperforms baselines across these tasks indicates that the observed advantages in MASBench **successfully translate to improved outcomes in realistic settings**.
> > >
> > > Regarding single-step orchestration, our claim is not that it fully resolves adaptability, but that **it is effective at producing strong initial MAS structures**. Empirically, even in long-horizon and tool-use settings, a single-step generated structure already captures effective coordination patterns. Our formulation of building the entire MAS in one step simplifies credit assignment in RL and avoids error propagation. Rather than building an MAS incrementally, one component at a time, our solution gives a competitive and effective alternative, where subsequent steps (not explored in this paper) can be simple edits to the initial MAS structure.
> > >
> > > For baseline comparison, we would like to clarify that prior work often **does not** include training-time baselines (e.g., [1,2,3,4]), potentially due to the difficulties we mentioned. Even when such comparisons are attempted (e.g., [5]), key factors such as **fixing the sub-agent space** or **evaluating on out-of-domain tasks** are not controlled. In our work, we explicitly address these challenges by introducing a unified evaluation protocol. While we agree that perfect comparability is difficult, we believe this represents a meaningful step toward more controlled and informative comparisons.
> > >
> > > [1]: MAS-GPT: Training LLMs to Build LLM-based Multi-Agent Systems. ICML 2025.
> > > [2]: ToolOrchestra: Elevating Intelligence via Efficient Model and Tool Orchestration, 2025.
> > > [3]: Controlling Performance and Budget of a Centralized Multi-agent LLM System with Reinforcement Learning, 2025.
> > > [4]: Weak-for-Strong: Training Weak Meta-Agent to Harness Strong Executors. COLM 2025.
> > > [5]: How to Train a Leader: Hierarchical Reasoning in Multi-Agent LLMs, 2025

---

### Official Review · Reviewer_gLqU · 2026-03-13

**Soundness:** 2
**Presentation:** 2
**Significance:** 3
**Originality:** 3
**Overall Recommendation:** 4
**Confidence:** 3

**Summary:**

This paper identifies two core limitations in current automatic multi-agent system design: methodological complexity (where sequential, code-level execution limits global system-level reasoning) and efficacy uncertainty (where systems are deployed without understanding if they offer tangible benefits over single-agent systems). To address these issues, the authors propose MAS-Orchestra, a training-time framework that formulates MAS orchestration as a function-calling reinforcement learning problem. This enables holistic orchestration, generating the entire MAS topology in a single step. Additionally, the authors introduce a highly valuable controlled benchmark, MASBENCH, which characterizes tasks along five axes (Depth, Horizon, Breadth, Parallel, and Robustness) to systematically analyze when and why MAS is beneficial. Experiments demonstrate that MAS-Orchestra achieves state-of-the-art performance on public benchmarks like math and multi-hop QA, while operating at significantly lower costs than strong baselines.

**Compliance With Llm Reviewing Policy:**

Affirmed.

**Final Justification:**

I keep my positive score

**Key Questions For Authors:**

See Weaknesses

**Limitations:**

Yes

**Strengths And Weaknesses:**

Strengths:
1. MAS-Orchestra encapsulates complex sub-agents as black-box callable functions and generates the complete orchestration in a single decision step. This significantly simplifies the optimization process and allows the orchestrator to focus on global system design.
2.  The proposed MASBENCH is a major contribution, providing a quantitative basis for the "MAS vs. SAS" debate by controlling the underlying dependency graph structures (e.g., dependency chain length, max in-degree) and injecting adversarial notes.
3. On challenging tasks like AIME and GPQA, MAS-Orchestra delivers higher or comparable accuracy while reducing inference calls and overall costs by more than 10x compared to similar methods, successfully achieving the performance-cost Pareto frontier.

Weaknesses:
1. The concept of DoM (low and high) is introduced to constrain the orchestration space, but it is explicitly configured by the user rather than dynamically learned and decided end-to-end by the orchestrator. This limits the framework's fully autonomous capability on unknown or mixed task distributions.
2. To answer what factors affect MAS performance, the candidate sub-agents were strictly limited to a single CoTAgent (excluding others like SearchAgent) to remove confounding effects. While this ensures rigorous variable control, it fails to capture the complex dynamics of heterogeneous agent collaboration that occur in real-world scenarios.
3. Appendix C.2 reveals that if adversarial samples are excluded from the combined training data, MAS performance on the Robustness axis remains poor, performing nearly as badly as an SAS. This demonstrates that the MAS architecture itself is not inherently immune to noise; its robustness advantage is highly dependent on specific adversarial-aware training.

---

> ### Author Rebuttal · Authors · 2026-03-31
>
> Thank you for the thorough review and for recognizing the simplicity and effectiveness of our method, as well as the strong contribution of MASBench for controlled evaluation of MAS vs. SAS! Please see our response to your comments/questions below.
>
> > W1: The concept of DoM is explicitly configured by the user rather than dynamically learned and decided end-to-end by the orchestrator
>
> Our MASBench provides empirical guidance on how to set DoM. Specifically, we find that MAS is most beneficial for tasks requiring parallelism or involving adversarial context. Based on this, we set different DoM for different benchmarks (L387-L400). That said, we agree that an ideal approach would be to learn or infer DoM automatically, which is an interesting direction for future work. As a first step, our work aims to systematically study when MAS is beneficial, and provide empirical insights to shed light on this important direction.
>
> > W2: To answer what factors affect MAS performance, the candidate sub-agents were strictly limited to a single CoTAgent (excluding others like SearchAgent) to remove confounding effects. While this ensures rigorous variable control, it fails to capture the complex dynamics of heterogeneous agent collaboration that occur in real-world scenarios.
>
> As the first benchmark studying when MAS is beneficial compared to SAS,  MASBench is designed to emphasize **controllability**. By restricting to a single CoTAgent, we ensure that performance differences arise from orchestration rather than heterogeneous agent capabilities.
>
> To capture more realistic multi-agent dynamics, we **complement** MASBench with public benchmarks (AIME, GPQA, HotpotQA, BrowseComp+), where heterogeneous agents are used and more complex interactions emerge. Our design principle is to separate **controlled analysis** and **realistic evaluation**: MASBench provides a synthetic and controlled setting for understanding when MAS helps, while public benchmarks reflect real-world scenarios with richer agent interactions.
>
> >W3: Appendix C.2 reveals that if adversarial samples are excluded from the combined training data, MAS performance on the Robustness axis remains poor, performing nearly as badly as an SAS. This demonstrates that the MAS architecture itself is not inherently immune to noise; its robustness advantage is highly dependent on specific adversarial-aware training.
>
> We agree with this observation. As shown in Appendix C.2, MAS performance on the robustness axis depends on learning an effective orchestration topology, which is enabled by adversarial-aware training. Without such training, MAS does not outperform SAS.
>
> We would like to clarify that this is not a weakness of our analysis, but rather an important empirical finding. A key goal of MASBench is to identify when MAS provides benefits over SAS. Our results show that robustness gains are not inherent to MAS, but arise when the system is trained to handle adversarial conditions. This indicates that MAS provides a flexible framework that can learn robustness when exposed to appropriate training signals. We view this as a useful insight for the community rather than a limitation.

---

> > ### Author Rebuttal · Reviewer_gLqU · 2026-04-03
> >
> > Thanks for authors' reply, it addressed most of my concerns, I would like to keep my positive score.

---

### Official Review · Reviewer_UWQo · 2026-03-17

**Soundness:** 3
**Presentation:** 3
**Significance:** 3
**Originality:** 4
**Overall Recommendation:** 4
**Confidence:** 4

**Summary:**

This paper studies a multi-agent reasoning system for large language models and proposes MAS-Orchestra, a training framework that automatically designs and coordinates multi-agent systems through holistic orchestration. The method formulates agent orchestration as a reinforcement learning problem, where complex sub-agents are abstracted as callable functions and a learned orchestrator generates the entire multi-agent structure, enabling global reasoning over agent roles and interactions.

**Compliance With Llm Reviewing Policy:**

Affirmed.

**Key Questions For Authors:**

1. In the evaluation, more explanation of the results can be provided. The performance on the latency, cost, and reliability can be added.
2. The RL model needs to be more comprehensive. What specific RL algorithm is adopted needs to be justified.
3. The implementation details of the proposed framework, such as orchestration parsing and training, can be better explained.

**Limitations:**

yes

**Strengths And Weaknesses:**

1. In general, the proposed framework is technically well defined. However, the reward design is relatively simple, which may not fully capture the quality of intermediate orchestration decisions. In the simulations, the reason why the proposed method works better needs to be explained instead of only showing the observation. The performance on the latency, cost, and reliability can be added in the simulations.
2. This paper is well structured. The implementation details, such as orchestration parsing and training, can be better explained.
3. The paper addresses an important aspect of multi-agent systems.
4. This paper seems novel.

---

> ### Author Rebuttal · Authors · 2026-03-31
>
> Thank you for the thorough review and for recognizing the novelty, technical soundness, clear structure, and importance of our work! Please see our response to your comments/questions below.
>
> > W1.1:  The reward design is relatively simple, which may not fully capture the quality of intermediate orchestration decisions.
>
> While task-level reward is simple, this design is intentional and empirically effective. Reward for intermediate orchestration decisions is difficult to define and often noisy (e.g., using an LLM judge for MAS remains an open problem). In preliminary experiments, we also explored augmenting the reward with orchestration-level signals (e.g., orchestration syntactic correctness, similar to format rewards used in reasoning tasks), but observed no consistent improvement. We therefore retain the final-answer reward.
>
> > W1.2: In the simulations, the reason why the proposed method works better needs to be explained instead of only showing the observation.
>
> The improvements can be attributed to three key design choices:
>
> **(1) Holistic orchestration.** Compared to sequential orchestration (ToolOrchestra) in Table 3, MAS-Orchestra achieves significantly better performance. The reason is that holistic orchestration generates the full MAS topology in one step, enabling globally coordinated designs. For instance, on BrowseComp+ (11.00 vs. 1.38), MAS-Orchestra learns to launch 3–4 parallel SearchAgents and route their outputs to an aggregation agent (Figure D.3), a pattern that sequential orchestration cannot discover, as it commits to agents one at a time and gets locked into locally suboptimal topologies (Table C.2).
>
> **(2) Function-calling RL.** Compared to MAS-GPT (which trains without function abstraction or RL) in Table 3, MAS-Orchestra again performs significantly better (e.g., 66.25 vs. 58.75 on AIME24). By encapsulating complex sub-agents as callable functions, the orchestrator focuses on high-level delegation rather than reproducing sub-agent internals. As shown in Figure D.1, MAS-Orchestra learns to dynamically select the strongest sub-agent per task (ReflexionAgent or DebateAgent), effectively learning *which* agent to use rather than committing to a fixed choice.
>
> **(3) Degree of MAS (DoM).** Our MASBench analysis (Section 5) shows that MAS gains depend on task structure: MAS excels in parallel and adversarial settings but provides limited benefit for purely sequential reasoning (Depth axis, Figure 2). This directly informs our DoM configuration, low DoM for reasoning tasks (AIME, GPQA) and high DoM for tasks with parallel sub-queries (HotpotQA, BrowseComp+), and the public benchmark results in Table 3 are consistent with these controlled findings.
>
>
> > W1.3, Q1: The performance on the latency, cost, and reliability can be added in the simulations.
>
> We report detailed latency (wall-time in ks) and cost (#LLM calls, tokens in M, and cost in $) in Appendix D.4, with a summary in Figure 1. MAS-Orchestra is substantially more efficient: up to **150×** lower cost than training-time orchestration, up to **100×** lower cost than inference-time methods, and up to 10× lower cost than standalone agents. It is also significantly faster (up to **250×** vs. strong baselines), with latency comparable to CoTAgent. For all the experiments (both MASBench and public benchmarks), we report the average results over **8** independent inference runs (avg@8), which is the largest number of independent runs across all the considered baselines, to ensure the reliability of our results.
>
> > W2, Q3: The implementation details, such as orchestration parsing and training, can be better explained.
>
> Most implementation details are provided in the appendix. Appendix B describes the training setup, including hyper-parameters, hardware requirement, RL framework and correctness computation. Appendix G and H detail the orchestrator prompts and the orchestration parser design. We will also open-source the full codebase, including models, training and evaluation pipelines, and MASBench, to support reproducibility and further exploration.
>
> > Q2: The RL model needs to be more comprehensive. What specific RL algorithm is adopted needs to be justified.
>
> In Section 3.2, We state that we use GRPO for RL training. We adopt GRPO because it naturally fits our **holistic orchestration** setting, where each sampled orchestration receives a task-level reward, and it is also a simple and standard choice in the current community. The details of GRPO are provided in Appendix F.

---

### Decision · Program_Chairs · 2026-04-30

**Decision:**

Accept (regular)

**Comment:**

Reviewers appreciated the important topic, the paper's clarity, the proposed framework's soundness and novelty, and the empirical benefits in terms of accuracy and cost. They particularly saw a potential for impact in the new and well-motivated benchmark, including the insightful DoM quantification. Reviewer concerns seem to have been mostly addressed in rebuttal, including issues of clarity and detail, concerns of limited evaluation, and several extensions that this work leaves out of scope (e.g. automating DoM prediction). Recurring concerns that have not been fully addressed include the possible downsides of single-step orchestration, such as lack of adaptability, and the challenges of fair baseline comparison; these should at least be more explicitly noted in the paper.